# Lipschitz Bounds and Provably Robust Training by Laplacian Smoothing

**Vishaal Krishnan**
Mechanical Engineering Department
University of California Riverside
vishaalk@ucr.edu

**Abed AlRahman Al Makdah**
Electrical & Computer Engineering Department
University of California Riverside
aalmakdah@engr.ucr.edu

**Fabio Pasqualetti**
Mechanical Engineering Department
University of California Riverside
fabiopas@engr.ucr.edu

## Abstract

In this work we propose a graph-based learning framework to train models with provable robustness to adversarial perturbations. In contrast to regularization-based approaches, we formulate the adversarially robust learning problem as one of loss minimization with a Lipschitz constraint, and show that the saddle point of the associated Lagrangian is characterized by a Poisson equation with weighted Laplace operator. Further, the weighting for the Laplace operator is given by the Lagrange multiplier for the Lipschitz constraint, which modulates the sensitivity of the minimizer to perturbations. We then design a provably robust training scheme using graph-based discretization of the input space and a primal-dual algorithm to converge to the Lagrangian's saddle point. Our analysis establishes a novel connection between elliptic operators with constraint-enforced weighting and adversarial learning. We also study the complementary problem of improving the robustness of minimizers with a margin on their loss, formulated as a loss-constrained minimization problem of the Lipschitz constant. We propose a technique to obtain robustified minimizers, and evaluate fundamental Lipschitz lower bounds by approaching Lipschitz constant minimization via a sequence of gradient $p$-norm minimization problems. Ultimately, our results show that, for a desired nominal performance, there exists a fundamental lower bound on the sensitivity to adversarial perturbations that depends only on the loss function and the data distribution, and that improvements in robustness beyond this bound can only be made at the expense of nominal performance. Our training schemes provably achieve these bounds both under constraints on performance and robustness.

## 1 Introduction

Sensitivity to adversarial perturbations is one of the main limitations of data-driven models, and a hurdle to their deployment in safety-critical applications. Improving adversarial robustness requires adjusting the worst-case sensitivity of the data-driven input-output map, which is characterized by its Lipschitz constant. Training under a Lipschitz regularization or constraint is therefore a natural way of improving adversarial robustness, which has led to many works on the subject [1, 2]. Yet, a fundamental understanding of the limitations of this approach, as well as a general framework for training models that are provably robust to adversarial perturbations, remain critically lacking.

Motivated by this need, we consider the problem of adversarially robust learning, formulated as a loss minimization problem with a Lipschitz constraint:

$$\inf_{f \in \mathrm{Lip}(\mathbb{X};\mathbb{Y})} \underbrace{\mathbb{E}_{(x,y) \sim \sigma} \left[ \ell \left( f(x), y \right) \right]}_{\triangleq L_\sigma(f)}, \qquad \text{s.t. } \mathrm{lip}(f) \leq \alpha, \tag{1}$$

where $\mathbb{X}$ and $\mathbb{Y}$ are the input and output spaces equipped with distance functions, $\ell$ is the loss function for the learning problem, $\sigma$ the data-generating distribution and the search space is the space $\mathrm{Lip}(\mathbb{X};\mathbb{Y})$ of Lipschitz-continuous maps from $\mathbb{X}$ to $\mathbb{Y}$ with an upper bound $\alpha$ on the Lipschitz constant. This class of problems includes, for instance, the problem of image classification with a constraint on the Lipschitz constant of the classifier. In this case, $x$ denotes an image, $y$ a probability vector over the space of labels and $\sigma$ captures the relation between images and labels. In (1), we do not restrict our attention to any finite-dimensional subspace of $\mathrm{Lip}(\mathbb{X};\mathbb{Y})$, as done when a particular machine learning model is chosen (for instance, neural network, where the dimension of the search space is specified by the network structure). Instead, we focus on the infinite-dimensional learning problem to derive insights and fundamental bounds for the underlying adversarial learning problem. Finally, imposing a hard constraint on the Lipschitz constant (as opposed to a regularization term) allows us to provide hard guarantees on the robustness of the minimizer to adversarial perturbations.

**Contributions.** In this paper we characterize fundamental robustness bounds for machine learning algorithms, and design provably robust training schemes. Our approach creates, to the best of our knowledge, a novel and useful bridge between the nascent theory of provably robust learning and the classic theories of elliptic operators, partial differential equations, and numerical integration. The technical contributions of this paper are twofold. First, in Section 2 we consider Problem (1) of designing a data-driven map to minimize the loss function, with a desired bound on the map's Lipschitz constant. Under assumptions on strict convexity of the loss function and compactness of the input and output spaces, we show that the problem has a unique minimizer and characterize the saddle point of the corresponding Lagrangian for the problem as the (weak) solution to a Poisson partial differential equation involving a weighted Laplace operator, with the weighting given by the Lagrange multiplier for the constraint. This result provides key insights into the nature of the optimal data-driven map satisfying robustness constraints. We then design a provably robust training scheme based on a graph discretization of the domain to numerically solve for the minimizer of the problem.

Second, we consider the problem of minimizing the Lipschitz constant of a data-driven map with a guaranteed bound (margin) on its loss. We show that the Lipschitz constant is tightly and inversely related to the loss, thereby revealing a fundamental tradeoff between the robustness of a data-driven map and its performance. This result implies that the Lipschitz contant of any data-driven algorithm achieving a desired level of performance has a fundamental lower bound that depends only on the loss function $\ell$ and the data-generating distribution $\sigma$, which constitutes a fundamental lower bound to benchmark any training algorithm and learning problem. We also provide a training scheme for further improving the robustness of a minimizer with a margin on the loss, by using a graph-based iterative procedure that involves solving a series of $p$-Poisson equations, decsribed in Section 3.

**Related work.** Motivated by real-world incidents and empirical studies [3], the issue of robustness of data-driven models to adversarial perturbations has received extensive attention in the last years [4–7]. When perturbations are chosen carefully, early studies [8] have shown that small input variations can cause large prediction errors in otherwise highly accurate neural networks. Several frameworks exist to design robust data-driven models, including regularization [1], adversarial training [9], distributionally robust optimization [10] and training under Lipschitz constraints. Of the above, the latter approach is particularly attractive, as it results in trained models with certified robustness.

The study of robustness of the class of neural network models has particularly drawn a lot of attention [11–17]. Many works [18–21] explore, in particular, the problem of training networks with Lipschitz constraints, and related issues. The complementary problem of estimating the Lipschitz constant of a trained neural network is also a crucial part of providing robustness certificates for trained models, and avoiding the danger of deploying unsafe models under a false sense of security. Recent works [22–24] have focused on deriving upper bounds on the Lipschitz constant of neural networks. While these certificates and training schemes provide a way of estimating and improving robustness of a certain class of data-driven models, they fall short in providing insight into the fundamental robustness bounds for the underlying learning problem and the means to exploit them in design.

Furthermore, recent works also point towards fundamental tradeoffs between accuracy and robustness of data-driven models [25–28] in various settings and training frameworks. The connection of adversarial robustness to model complexity and generalization, and the existence (or non-existence) of fundamental tradeoffs between them is another important problem that has received attention [29–34], and is the subject of ongoing debate. This paper builds and extends upon these early studies.

**Notation.** We introduce here some useful notation. We use $|\cdot|$ to denote the Euclidean norm in $\mathbb{R}^d$, for any $d \in \mathbb{N}$ (when $d = 1$, this denotes the absolute value) and more generally the Hilbert-Schmidt (H-S) norm in finite dimensions. We use $\|\cdot\|$ for function space norms. For maps $f$ between high-dimensional spaces, we often require the notation $\||f|\|$, which specifies the function space norm of $|f|$ (which is in turn the function that evaluates to the H-S norm of the map $f$ at any point in its domain). For $\mathbb{X} \subset \mathbb{R}^{\dim(\mathbb{X})}$, we denote by $(\mathbb{X}, \mu)$ the set $\mathbb{X}$ with an underlying measure $\mu$. We denote by $\mathcal{F}(\mathbb{X}; \mathbb{Y})$ a class $\mathcal{F}$ (placeholder for the particular spaces mentioned below) of maps from $\mathbb{X}$ to $\mathbb{Y}$. We denote by $L^p(\mathbb{X}, \mu)$ the space of $p$-integrable (measurable) functions on $\mathbb{X}$, where the integration is carried out with the underlying measure $\mu$ (the Lebesgue measure is implied when $\mu$ is not specified), and by $W^{1,p}(\mathbb{X}, \mu)$ the space of $p$-integrable (measurable) functions with $p$-integrable (measurable) derivatives. When generalized to the space of maps, as in $f \in L^p((\mathbb{X}, \mu); \mathbb{Y})$, we mean $|f| \in L^p(\mathbb{X}, \mu)$. Also, for $f \in W^p((\mathbb{X}, \mu); \mathbb{Y})$, we mean $|f| \in L^p(\mathbb{X}, \mu)$ and $|\nabla f| \in L^p(\mathbb{X}, \mu)$.

## 2 Lipschitz-constrained loss minimization and provably robust training

In this section we study and solve the Lipschitz constrained loss minimization problem (1). We start by specifying the setting for Problem (1). Let $\mathbb{X} \subset \mathbb{R}^{\dim(\mathbb{X})}$ and $\mathbb{Y} \subset \mathbb{R}^{\dim(\mathbb{Y})}$ be convex and compact, $\sigma$ an absolutely continuous probability measure on $\mathbb{X} \times \mathbb{Y}$ with (absolutely continuous) marginal $\mu$ supported on $\mathbb{X}$ and conditional $\pi$. Let the loss function $\ell : \mathbb{Y} \times \mathbb{Y} \to \mathbb{R}_{\geq 0}$ be strictly convex and Lipschitz continuous. The Lipschitz constraint on the maps in (1) is a global constraint involving every pair of points in the domain $\mathbb{X}$. To obtain a tractable formulation, we equivalently rewrite the Lipschitz constraint as a bound on the norm of the gradient in the domain $\mathbb{X}$. The space of Lipschitz continuous maps $\mathrm{Lip}(\mathbb{X}; \mathbb{Y})$ is also the Sobolev space $W^{1,\infty}((\mathbb{X}, \mu); \mathbb{Y})$ of essentially bounded (measurable) maps with essentially bounded (measurable) gradients, that is, $\mathrm{Lip}(\mathbb{X}; \mathbb{Y}) = W^{1,\infty}((\mathbb{X}, \mu); \mathbb{Y})$.[1] The Lipschitz constant of a map $f \in \mathrm{Lip}(\mathbb{X}; \mathbb{Y})$ is $\mathrm{lip}(f) = \||\nabla f|\|_{L^\infty((\mathbb{X}, \mu); \mathbb{Y})}$ (the $W^{1,\infty}$-seminorm of $f$). We refer the reader to our supplementary material or [35] for a discussion of these notions.

Using the above definitions, the Lipschitz constrained loss minimization problem (1) becomes

$$\inf_{f \in W^{1,\infty}((\mathbb{X}, \mu); \mathbb{Y})} \left\{ L_\sigma(f), \quad \text{s.t.} \quad \||\nabla f|\|_{L^\infty(\mathbb{X}, \mu)} \leq \alpha \right\}. \tag{2}$$

To see the role of the Lipschitz constant in the sensitivity of the loss to adversarial perturbations, first notice that adversarial perturbations can be written as the perturbations on the joint distribution $\sigma$ generated by a map $T$ that perturbs the inputs $x \in \mathbb{X}$ while preserving the outputs $y \in \mathbb{Y}$ [8]. In compact form, the class of adversarial perturbations can be written as:

$$\mathcal{T} = \left\{ T \mid T(x, y) = (T_1(x, y) , y), \text{ s.t. } T_1(x, y) \in B_\delta(x) \cap \mathbb{X} \right\},$$

where $B_\delta(x)$ is the open ball in $\mathbb{R}^{\dim(\mathbb{X})}$ of radius $\delta > 0$ and centered at $x$. Defining the sensitivity as the worst-case increase of the loss $L_\sigma$ following an adversarial perturbation $T \in \mathcal{T}$ for any $\sigma$, we get[2] that it is modulated by $L^\infty$-norm of the gradient $\nabla_1 \ell \cdot \nabla f$ (precisely, $\||\nabla_1 \ell \cdot \nabla f|\|_{L^\infty(\mathbb{X} \times \mathbb{Y}, \sigma)}$)[3] and whose upper bound is determined by the Lipschitz constant:

$$\underbrace{\||\nabla_1 \ell \cdot \nabla f|\|_{L^\infty(\mathbb{X} \times \mathbb{Y}, \sigma)}}_{\text{sensitivity of } L \text{ to adv. perturbation}} \leq \underbrace{\||\nabla_1 \ell|\|_{L^\infty(\mathbb{X} \times \mathbb{Y}, \sigma)}}_{\text{Lipschitz constant of } \ell} \cdot \underbrace{\||\nabla f|\|_{L^\infty(\mathbb{X}, \mu)}}_{\text{Lipschitz constant of } f} . \tag{3}$$

Problem 2 is convex (owing to the strict convexity of the loss $L_\sigma$[4] and the convexity of the constraint). Thus, we can expect to obtain a (unique) minimizer from the saddle point of the corresponding Lagrangian. With $G_f(x) = \frac{1}{2} \left( |\nabla f(x)|^2 - \alpha^2 \right)$, we can reformulate the Lipschitz constraint as

$G_f \leq 0$ $\mu$–a.e. in $\mathbb{X}$[5]. Since $f \in W^{1,\infty}((\mathbb{X}, \mu); \mathbb{Y})$, the constraint function $G_f$ belongs to the space $L^\infty(\mathbb{X}, \mu)$. Correspondingly, the Lagrange multiplier for the constraint $G_f \leq 0$ ($\mu$–a.e. in $\mathbb{X}$) is non-negative[6] and belongs to the dual space of $L^\infty(\mathbb{X}, \mu)$, which we denote as $\lambda \in L^\infty(\mathbb{X}, \mu)^*_{\geq 0}$. The Lagrangian $\mathcal{L}_\sigma : W^{1,\infty}((\mathbb{X}, \mu); \mathbb{Y}) \times L^\infty(\mathbb{X}, \mu)^*_{\geq 0}$ for Problem (2) is then given by:

$$\mathcal{L}_\sigma(f, \lambda) = L_\sigma(f) + \lambda(G_f). \tag{4}$$

**Theorem 2.1.** *(Lipschitz constrained loss minimization) Problem (2) has a unique global minimizer $f^* \in W^{1,\infty}((\mathbb{X}, \mu); \mathbb{Y})$. The Lagrangian $\mathcal{L}_\sigma$ has a unique saddle point $(f^*, \lambda^*) \in W^{1,\infty}((\mathbb{X}, \mu); \mathbb{Y}) \times L^1(\mathbb{X}, \mu)_{\geq 0}$. Moreover, $(f^*, \lambda^*)$ satisfies the first-order optimality conditions:*

1. *Stationarity: The saddle point $(f^*, \lambda^*)$ is a weak solution of the Poisson equation,*

$$-\frac{1}{\mu}\nabla \cdot (\mu\lambda^*\nabla f^*) + g_{f^*} = 0 \quad in\ \mathbb{X}, \qquad \mu\lambda^*\nabla f^* \cdot \mathbf{n} = 0 \quad on\ \partial\mathbb{X}, \tag{5}$$

   *where $g_{f^*}(x) = \mathbb{E}_{y \sim \pi(y \mid x)}[\nabla_1 \ell(f^*(x), y)]$ and $\mathbf{n}$ is the outward normal to the boundary $\partial\mathbb{X}$.*

2. *Feasibility: $|\nabla f^*| \leq \alpha$ and $\lambda^* \geq 0$, $\mu - a.e.$ in $\mathbb{X}$.*

3. *Complementary slackness: $\lambda^*(|\nabla f^*| - \alpha) = 0$, $\mu - a.e.$ in $\mathbb{X}$.*

Some comments on Theorem 2.1 are in order. In the absence of the constraint in (2) (that is, $\alpha = \infty$), the stationarity condition is characterized by $\mathbb{E}_{y \sim \pi(y \mid x)}[\nabla_1 \ell(f^*_{\mathrm{unc}}(x), y)] = 0$, where $f^*_{\mathrm{unc}}(x), y)$ is the unconstrained minimizer of the loss functional. The saddle point of $\mathcal{L}_\sigma$ is characterized by the Poisson equation (5), which encodes the stationarity condition for the Lagrangian. The Neumann boundary condition in (5) results from the fact that we do not enforce a boundary constraint on the map in the loss minimization problem (2). The $\lambda^*$-weighted Laplace operator, $\frac{1}{\mu}\nabla \cdot (\mu\lambda^*\nabla)$, is responsible for locally enforcing the Lipschitz constraint and regularizing (smoothing) the minimizer. Moreover, the Lagrange multiplier satisfies $\lambda^* \in L^1(\mathbb{X}, \mu)_{\geq 0}$, and is therefore integrable (this is stronger regularity than in the definition $\lambda \in L^\infty(\mathbb{X}, \mu)^*_{\geq 0}$). It follows from the feasibility condition in Theorem 2.1 that the minimizer (provably) satisfies the Lipschitz bound (in contrast to Lipschitz regularization-based approaches to adversarial learning). From the complementary slackness condition in Theorem 2.1, smoothing is enforced only when the constraint is active: when the constraint is inactive in a region $D \subset \mathbb{X}$ of non-zero measure (that is, $|\nabla f^*(x)| < \alpha$ for $x \in D$ and $\mu(D) > 0$), the Lagrange multiplier satisfies $\lambda^* = 0$ ($\mu$-a.e. in $D$) and smoothing is not enforced.

The fact that the saddle point of the Lagrangian $\mathcal{L}_\sigma$ in (4) satisfies the Lipschitz bound forms the basis for the design of a provably robust training scheme, which we obtain through a discretization of Problem (2) over a graph. To this end, we select $n$ points $\{X_i\}_{i=1}^n$, $X_i \in \mathbb{X}$, via i.i.d. sampling of the distribution $\mu$ (in practice, we sample uniformly i.i.d. from the input dataset, that defines the empirical marginal measure $\widehat{\mu}$). With the discretization points $\{X_i\}_{i=1}^n$ as the (embedding of) vertices, we construct an undirected, weighted, connected graph $\mathcal{G} = (\mathcal{V}, \mathcal{E}, W)$, with vertex set $\mathcal{V} = \{1, \ldots, n\}$, edge set $\mathcal{E} = \mathcal{V} \times \mathcal{V}$, and weighted adjacency matrix $W = [w_{ij}]_{i,j=1}^n$.

We assume the availability of a labeled dataset $D = \{(x_i, y_i)\}_{i=1}^N$ consisting of $N > n$ i.i.d. samples of $\sigma$, and define a partition $\mathcal{W} = \{\mathcal{W}_i\}_{i=1}^n$ of the dataset $D$ as follows:

$$\mathcal{W}_i = \{(x, y) \in D \mid |x - X_i| \leq |x - X_j| \ \forall\ j \in \mathcal{V} \setminus \{i\}\}. \tag{6}$$

We then assign weights $\theta_{ij} = N^{-1}$ to the samples $\xi_j = (x_j, y_j) \in \mathcal{W}_i$ (a different weighing scheme may affect generalization and performance of our model; we leave this for future research). Finally, we write the discrete (empirical) Lipschitz constrained loss minimization problem over the graph $\mathcal{G}$ as follows (this minimization problem can be viewed as the discretized version of (2) over $\mathcal{G}$):

$$\min_{\substack{\mathbf{v} = (v_1, \ldots, v_n) \\ v_i \in \mathbb{R}^{\dim(\mathbb{Y})}}} \left\{ \sum_{i \in \mathcal{V}} \left( \sum_{j \in \mathcal{W}_i} \theta_{ij}\ell(v_i, y_j) \right), \quad \text{s.t.}\ \ |v_r - v_s| \leq \alpha |X_r - X_s|, \ \forall\ (r, s) \in \mathcal{E} \right\}. \tag{7}$$

We note that the above constrained minimization problem (7) is convex (strictly convex objective function with convex constraints) and the corresponding Lagrangian is given by:

$$\mathcal{L}_{\mathcal{G}}(\mathbf{v}, \Lambda) = \sum_{i \in \mathcal{V}} \left[ \sum_{s \in \mathcal{W}_i} \theta_{is} \ell(v_i, y_s) + \frac{1}{2} \sum_{j \in \mathcal{V}} \lambda_{ij} w_{ij} \left( |v_i - v_j|^2 - \alpha |X_i - X_j|^2 \right) \right], \qquad (8)$$

where $\Lambda = [\lambda_{ij}]_{i,j=1}^n$ is the matrix of Lagrange multiplier for the pairwise Lipschitz constraints. Define a primal-dual dynamics for the Lagrangian $\mathcal{L}_{\mathcal{G}}(\mathbf{v}, \Lambda)$ with time-step sequence $\{h(k)\}_{k \in \mathbb{N}}$:

$$\begin{aligned} \mathbf{v}(k+1) &= \mathbf{v}(k) - h(k) \, \nabla_{\mathbf{v}} \mathcal{L}_{\mathcal{G}} \left( \mathbf{v}(k), \Lambda(k) \right), \\ \Lambda(k+1) &= \max\{0 \, , \, \Lambda(k) + h(k) \, \nabla_{\Lambda} \mathcal{L}_{\mathcal{G}} \left( \mathbf{v}(k), \Lambda(k) \right)\}. \end{aligned} \qquad (9)$$

The primal dynamics is a discretized heat flow over the graph $\mathcal{G}$ with a weighted Laplacian, where $\nabla_{\mathbf{v}} \mathcal{L}_{\mathcal{G}} \left( \mathbf{v}(k), \Lambda(k) \right) = \left( \Delta(\Lambda, W) \otimes I_{\dim(\mathbb{Y})} \right) \mathbf{v} + \theta \cdot \nabla_1 \ell(\mathbf{v}, \mathbf{y})$, and $\Delta(\Lambda, W)$ is the $\Lambda \circ W$-weighted Laplacian of the graph $\mathcal{G}$ (where $\circ$ denotes the Hadamard or entry-wise product of matrices). The convergence of the solution $\{(\mathbf{v}(k), \Lambda(k))\}_{k \in \mathbb{N}}$ of the primal-dual dynamics (9) to the saddle point of the Lagrangian $\mathcal{L}_{\mathcal{G}}$ follows [36] from the convexity of Problem (7).

As the size of the dataset $N$ and the size of graph $n$ increase, the solution to Problem (7) approaches the solution to Problem (2), under certain mild conditions. In particular, by the Glivenko-Cantelli Theorem [37], the empirical measure $\widehat{\sigma}_N = \frac{1}{N} \sum_{i=1}^N \delta_{(x_i, y_i)}$ converges uniformly and almost surely to the distribution $\sigma$ in the limit for $N \to \infty$, and so does $\widehat{\mu}_n = \frac{1}{n} \sum_{i=1}^n \delta_{X_i} \to \mu$ as $n \to \infty$, where $\delta$ here denotes the Dirac measure. Further, the convergence as $n \to \infty$ (higher model complexity) and $N \to \infty$ (larger dataset) of the minimizer of the (empirical) discrete minimization problem (7) to the infinite-dimensional problem (2) is modulated by the weights $\theta$ (which govern the convergence of the empirical loss) and $w$ (which governs the convergence of the graph Laplacian to the Laplace operator on the domain [38]).

We conclude this section with an illustrative example. Consider a dataset of $10000$ i.i.d. samples $(x_i, y_i)$, with $x_i \in [0, 1]^2$ and $y_i \in \{[1 \ 0]^\mathsf{T}, [0 \ 1]^\mathsf{T}\}$, taken uniformly from the distribution $\sigma$ in Fig. 1(a), where $y_i = [1 \ 0]^\mathsf{T}$ if $x_i$ belongs to a white cell and $y_i = [0 \ 1]^\mathsf{T}$ if $x_i$ belongs to a black cell. We randomly select $n$ nodes in $[0, 1]^2$, with $n = \{125, 200, 500\}$, construct a graph $\mathcal{G} = (\mathcal{V}, \mathcal{E})$ by connecting each node to its 10 nearest neighbors, and compute the solution $\mathbf{v}^*$ to (9) for different values of the Lipschitz constant $\alpha$. Then, we generate a testing set of $2000$ i.i.d. samples from $\sigma$, associate them with the closest node, and evaluate the classification confidence of $\mathbf{v}^*$. In particular, if the testing sample $\bar{x}_i$ is closest to the $i$-th node and $v_i^* = [p_1 \ p_2]^\mathsf{T}$, then $\bar{x}_i$ is classified as $[1 \ 0]^\mathsf{T}$ with confidence $p_1$ if $p_1 > p_2$, and as $[0 \ 1]^\mathsf{T}$ with confidence $1 - p_1$ if $p_1 < p_2$. Fig. 1(b)-(h) shows the Voronoi cells associated with the nodes $\mathcal{V}$, where each cell is colored on a gray scale using the first entries of $v_i^*$ (darker colors indicate higher confidence in classifying the samples in a cell as $[0 \ 1]^\mathsf{T}$, while lighter colors indicate higher confidence in classifying the samples in a cell as $[1 \ 0]^\mathsf{T}$). It can be seen that the classification confidence increases with the number of nodes and the Lipschitz bound, at the expenses of a higher model complexity and sensitivity to adversarial perturbations. This trend is also visible in Fig.1(i), where the classification confidence increases with the Lipschitz bound until it saturates for the classifier with highest confidence given the training set and discretization points.

## 3 Robustification with loss margin and fundamental bound

In this section we study the problem of increasing the robustness of a minimizer with a margin on the loss. Let $f^*$ be the minimizer of (1) with Lipschitz bound $\alpha$, and let $J_\sigma^*(\alpha)$ be the optimal loss. We formulate and solve the following loss constrained Lipschitz constant minimization problem:

$$\inf_{f \in W^{1,\infty}((\mathbb{X}, \mu); \mathbb{Y})} \left\{ \||\nabla f|\|_{L^\infty((\mathbb{X}, \mu); \mathbb{Y})}, \qquad \text{s.t. } L_\sigma(f) \le J_\sigma^*(\alpha) + \epsilon \right\}. \qquad (10)$$

Because the Lipschitz constant satisfies $\||\nabla f|\|_{L^\infty((\mathbb{X}, \mu); \mathbb{Y})} = \operatorname{ess\,sup} |\nabla f|$, Problem (10) has a $\min - \max$ (more precisely, an $\inf - \operatorname{ess\,sup}$) structure which is not amenable to tractable numerical schemes. We circumvent this hurdle by approaching problem (10) via a sequence of loss-constrained (convex) minimization problems involving the $W^{1,p}$-seminorm, for $p \in \mathbb{N}$, $p > 1$, given by:

$$\inf_{f \in W^{1,p}((\mathbb{X}, \mu); \mathbb{Y})} \left\{ \||\nabla f|\|_{L^p(\mathbb{X}, \mu)}, \qquad \text{s.t. } L_\sigma(f) \le J_\sigma^*(\alpha) + \epsilon \right\}. \qquad (11)$$

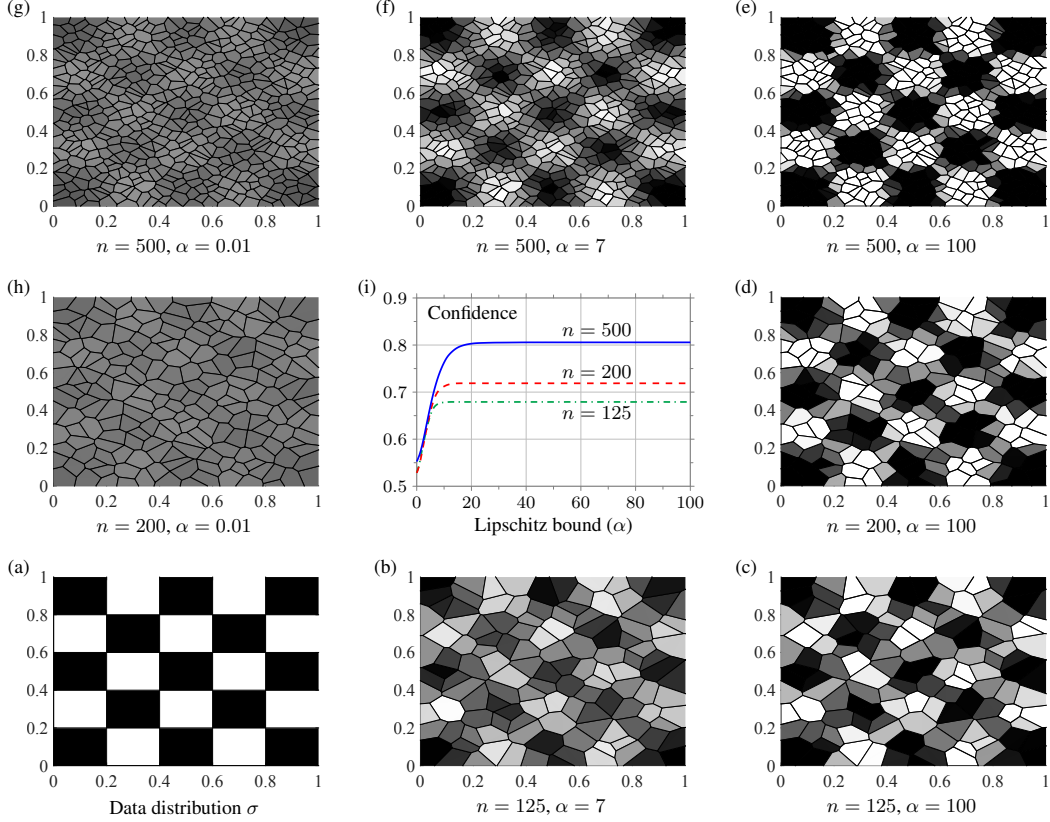

Figure 1: For the classification problem discussed in Section 2, this figure shows a tradeoff between the confidence of classification, the Lipschitz constant, and the complexity of the classifier designed using our algorithm (9). Increasing the Lipschitz constant of the classifier and its complexity also increases the confidence of classification, at the expenses of a higher sensitivity to perturbations.

$W^{1,p}$-seminorm minimization problems are typically formulated to obtain minimum Lipschitz extensions in semi-supervised learning [39–42]. A related problem is the one of $W^{1,p}$-seminorm regularized learning [43, 44]. Instead, we propose this approach, for the first time, to improve the robustness of minimizers to adversarial perturbations with a guaranteed margin on the loss.

Convexity of Problem (11) follows from the convexity of the $W^{1,p}$-seminorm in $W^{1,p}((\mathbb{X}, \mu); \mathbb{Y})$ and the strict convexity of $L_\sigma$ (which yields a convex constraint). The minimizers are obtained from the saddle points of the Lagrangian $\mathcal{H}_\sigma^p : W^{1,p}((\mathbb{X}, \mu); \mathbb{Y}) \times \mathbb{R}_{\geq 0} \to \mathbb{R}$ for Problem (11), given by:

$$\mathcal{H}_\sigma^p(f, \kappa) = \frac{1}{p} \, \||\nabla f|\|_{L^p(\mathbb{X}, \mu)}^p + \kappa \left( L_\sigma(f) - (J_\sigma^*(\alpha) + \epsilon) \right), \tag{12}$$

where we (equivalently) consider the $p$-th exponent $\||\nabla f|\|_{L^p(\mathbb{X}, \mu)}^p$ of the $W^{1,p}$-seminorm in defining the Lagrangian. The saddle points of $\mathcal{H}_\sigma^p$ are now specified by a Poisson equation involving the $p$-Laplace operator,[7] as established in the following theorem:

**Theorem 3.1.** *(Loss constrained $W^{1,p}$-seminorm minimization) For every $p \in \mathbb{N}_{>1}$, there exists a global minimizer $f^{\epsilon,p} \in W^{1,p}((\mathbb{X}, \mu); \mathbb{Y})$ for Problem (11). Also, there exists a saddle point $(f^{\epsilon,p}, \kappa^{\epsilon,p}) \in W^{1,p}((\mathbb{X}, \mu); \mathbb{Y}) \times \mathbb{R}_{\geq 0}$ of the Lagrangian $\mathcal{H}_\sigma^p$. Moreover, $(u, \kappa) \in W^{1,p}((\mathbb{X}, \mu); \mathbb{Y}) \times \mathbb{R}_{\geq 0}$ is a saddle point of $\mathcal{H}_\sigma^p$ if and only if it satisfies the following first-order optimality conditions:*

1. *Stationarity:* $(u, \kappa)$ *is a (weak) solution of the p-Poisson equation:*

$$-\Delta_p^\mu u + \kappa g_u = 0 \ \ in \ \mathbb{X}, \qquad \mu \nabla u \cdot \mathbf{n} = 0 \ \ on \ \partial \mathbb{X}, \tag{13}$$

   *where $g_u(x) = \mathbb{E}_{y \sim \pi(y \mid x)} [\nabla_1 \ell(u(x), y)]$ and $\Delta_p^\mu$ is the p-Laplace operator on $(\mathbb{X}, \mu)$.*

2. *Feasibility:* $L_\sigma(u) \leq J_\sigma^*(\alpha) + \epsilon$ *and* $\kappa \geq 0$.

3. *Complementary slackness:* $\kappa \left( L_\sigma(f) - (J_\sigma^*(\alpha) + \epsilon) \right) = 0$.

With the characterization of the minimizers of (11) for every $p \in \mathbb{N}$, $p > 1$ from Theorem 3.1, we now investigate whether the minimum value of (11) and its minimizers converge (as $p \to \infty$) to those of (10). The following theorem establishes that this is indeed the case, and that the minimum Lipschitz constant in (10) can be obtained as the limit of the sequence of minimum values of (11).

**Theorem 3.2.** *(Limit as $p \to \infty$ and fundamental Lipschitz lower bound) For any $\epsilon > 0$, it holds*

$$\lim_{p \to \infty} \min_{\substack{f \in W^{1,p}((\mathbb{X},\mu);\mathbb{Y}) \\ L_\sigma(f) \leq J_\sigma^*(\alpha) + \epsilon}} |\!|\!| \nabla f |\!|\!|_{L^p(\mathbb{X},\mu)} = \min_{\substack{f \in W^{1,\infty}((\mathbb{X},\mu);\mathbb{Y}) \\ L_\sigma(f) \leq J_\sigma^*(\alpha) + \epsilon}} |\!|\!| \nabla f |\!|\!|_{L^\infty((\mathbb{X},\mu);\mathbb{Y})}.$$

*Moreover, as $p \to \infty$, the sequence $\{f^{\epsilon,p}\}_{p \in \mathbb{N}_{>1}}$ of minimizers of Problem (11) converges uniformly to a (global) minimizer $f^{\epsilon,\infty}$ of (10).*

The facts that the saddle points of $\mathcal{H}_\sigma^p$ in (12) satisfy the bound on the loss (for every $p \in \mathbb{N}_{>1}$) for a given margin $\epsilon > 0$, and that the minimum value and minimizers of (11) converge in the limit $p \to \infty$ to those of (10), form the basis for the design of a robustification scheme. With the same graph structure and dataset partitioning as in Section 2, we write the discrete (empirical) loss-constrained $W^{1,p}$-seminorm minimization problem over the graph $\mathcal{G}$ as follows (this minimization problem can be viewed as the discretized version of (11) over the structure imposed by $\mathcal{G}$):

$$\min_{\substack{\mathbf{v}=(v_1,\ldots,v_n) \\ v_i \in \mathbb{R}^{\dim(\mathbb{Y})}}} \left\{ \frac{1}{p} \sum_{i \in \mathcal{V}} \sum_{j \in \mathcal{N}_i} w_{ij} |v_i - v_j|^p, \qquad \text{s.t.} \ \sum_{i \in \mathcal{V}} \sum_{s \in \mathcal{W}_i} \theta_{is} \ell(v_i, y_s) \leq J_\sigma^*(\alpha) + \epsilon \right\}. \tag{14}$$

We note that the above constrained minimization problem (14) is convex (convex objective function with convex constraints), and that the corresponding Lagrangian is given by:

$$\mathcal{H}_\mathcal{G}^p(\mathbf{v}, \kappa) = \sum_{i \in \mathcal{V}} \left[ \frac{1}{p} \sum_{j \in \mathcal{N}_i} w_{ij} |v_i - v_j|^p + \kappa \sum_{s \in \mathcal{W}_i} \left( \theta_{is} \ell(v_i, y_s) - \frac{1}{n}(J_\sigma^*(\alpha) + \epsilon) \right) \right], \tag{15}$$

The saddle points of (15) can be obtained via a primal-dual algorithm similar to (9) in Section 2. We solve the (discrete) loss-constrained Lipschitz minimization problem using an iterative procedure that employs the primal-dual algorithm to converge to a saddle point of $\mathcal{H}_\mathcal{G}^p$ in (15) at every iteration step $p \in \mathbb{N}_{>1}$. We then use the saddle point of $\mathcal{H}_\mathcal{G}^p$ as the initialization for the iteration step $p + 1$.

Theorem 3.1 offers key insights on the fundamental tradeoff between robustness and nominal performance. From complementary slackness in Theorem 3.1, it follows that, for the saddle points $(f^{\epsilon,p}, \kappa^{\epsilon,p})$, either the Lagrange multiplier satisfies $\kappa^{\epsilon,p} = 0$ or the constraint is active ($f^{\epsilon,p}$ occurs at the boundary of the constraint and the loss is $L_\sigma(f^{\epsilon,p}) = J^*(\alpha) + \varepsilon$). If the Lagrange multiplier is zero, then the Poisson equation characterizing the Stationarity condition (13) reduces to the $p$-Laplace equation with a Neumann boundary condition, whose solution is a constant map (in the weak sense). However, in practically useful cases (for small values of $\alpha$ and $\varepsilon$ with a low optimal loss $J^*(\alpha)$), there will typically not exist a constant map satisfying the loss margin $\varepsilon$ (unless the unconstrained minimizer $f_{\text{unc}}^*$ is itself flat). This implies that the Lagrange multiplier $\kappa$ is typically nonzero, that the minimizer $f^{\epsilon,p}$ occurs at the constraint boundary, and that the loss satisfies $L_\sigma((f^{\epsilon,p}) = J^*(\alpha) + \varepsilon$. Therefore, for every $p \in \mathbb{N}_{>1}$, the minimization problem (11) is typically dominated by the constraint, and the minimum value of the $W^{1,p}$-norm decreases monotonically with the loss margin. Thus, a fundamental tradeoff exists between performance and robustness.

We conclude this section with an example. Consider the classification problem described in Section 2. Fig. 2 shows the properties of the minimizers to (14) for varying values of $p$ and $\varepsilon$. It can be seen that, (i) as $p$ increases, the minimum value of (14) converges to its supremum value, which,

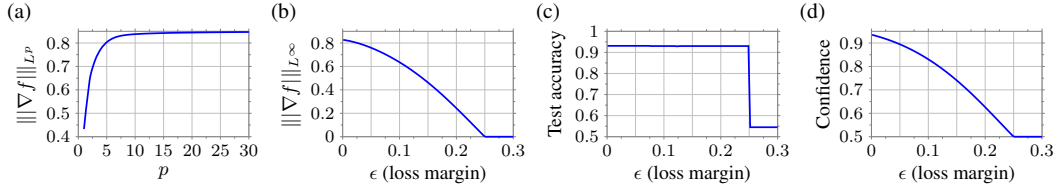

Figure 2: For the classification problem discussed in Section 2, (a) shows the convergence of the minimum values of (14) as $p \to \infty$ to the minimum Lipschitz constant (b) shows the tradeoff between performance and robustness (monotonic decrease of the minimum Lipschitz constant as a function of the loss margin), and (c)-(d) show the dependence of accuracy and confidence in testing on the loss margin obtained by solving (14) for different values of $\varepsilon$.

by Theorem 3.2, is smallest Lipschitz constant for a guaranteed loss margin $\varepsilon$ (Fig. 2(a)), and (ii) the minimum Lipschitz constant associated with the loss-constrained minimization problem is a monotonically non-increasing function of the loss margin $\varepsilon$, and strictly decreasing for small values of $\alpha$ and $\varepsilon$ (Fig. 2(b)). This curve describes a fundamental tradeoff between adversarial robustness and performance, and is entirely determined by the properties of the classification problem and not by the structure of the classifier. Fig. 2(c) and (d) show the dependence of accuracy and confidence in testing on the loss margin in training, and as expected, they are decreasing functions of the loss margin. We observe in (c) that the accuracy is constant at $0.93$ for $\epsilon < 0.25$ then drops to $0.54$ at $\epsilon = 0.25$. On the other hand, we observe that the confidence in (d) decreases smoothly with the loss margin for $\epsilon < 0.25$ till it reaches $0.5$ at $\epsilon = 0.25$. This implies that although the testing accuracy of the classifier remains at $0.93$ for $\epsilon < 0.25$, the classification is made with progressively lower confidence. For $\epsilon \geq 0.25$, the accuracy of the classifier is $0.54$ while the classification is made with a confidence of $0.5$ for each of the two classes.

## 4   Numerical experiments on MNIST dataset

In this section[8], we present the results from numerical experiments on the standard MNIST dataset of handwritten digits [45], for the training schemes in Sections 2 and 3. We first obtain $n = 5000$ graph vertices using the K-means algorithm on the images in the MNIST dataset. We then construct a graph $\mathcal{G} = (\mathcal{V}, \mathcal{E})$ by connecting each vertex to its 5 nearest neighbors, and compute the solution $\mathbf{v}^*$ to (9) for different values of the Lipschitz bound $\alpha$. We associate each testing data sample with the closest vertex, evaluate the classification confidence of $\mathbf{v}^*$, and assign to it the class that corresponds to the largest confidence. Fig. 3(a)-(c) show the dependence of testing accuracy, testing confidence, and testing loss on the Lipschitz bound $\alpha$. It can be seen that both accuracy and confidence increase with the Lipschitz bound, while the testing loss decreases with the Lipschitz bound. Fig 3(d) shows the relationship between the classifier's Lipschitz constant and the Lipschitz bound $\alpha$. It can be seen that the constraint in (2) is active for $\alpha < 175$, and inactive otherwise. Fig. 3(e) shows the dependence of the classifier's sensitivity to bounded perturbations, on the Lipschitz bound $\alpha$. The sensitivity of the trained classifier is the norm of the difference between the nominal and the perturbed confidence (confidence degradation). We observe that the sensitivity increases with the Lipschitz bound. Next, we fix the Lipschitz bound at $\alpha = 300$ and vary the complexity of the classifier by changing the number of vertices. We observe in Fig. 3(f)-(h) that the testing accuracy increases and the loss decreases with the number of vertices (model complexity), while the confidence remains almost constant. Finally, we fix the number of vertices at $n = 5000$ and compute the solution $\mathbf{v}^*$ to (14) for different values of the loss margin $\epsilon$. Fig. 3(i)-(k) show the dependence of the Lipschitz constant, testing accuracy and confidence on the loss margin. As predicted by our theory, and in accordance with the results obtained in the other numerical examples in Fig. 3(a), the classifier's Lipschitz constant (Fig. 3(i)), accuracy and confidence (Fig. 3(j),(k)) are decreasing functions of the classifier's loss margin. On the other hand, the model Lipschitz constant is directly proportional to the classification confidence (Fig. 3(l)). This confirms the existence of a tradeoff between robustness and performance, and provides a limiting benchmark for comparison with other models.

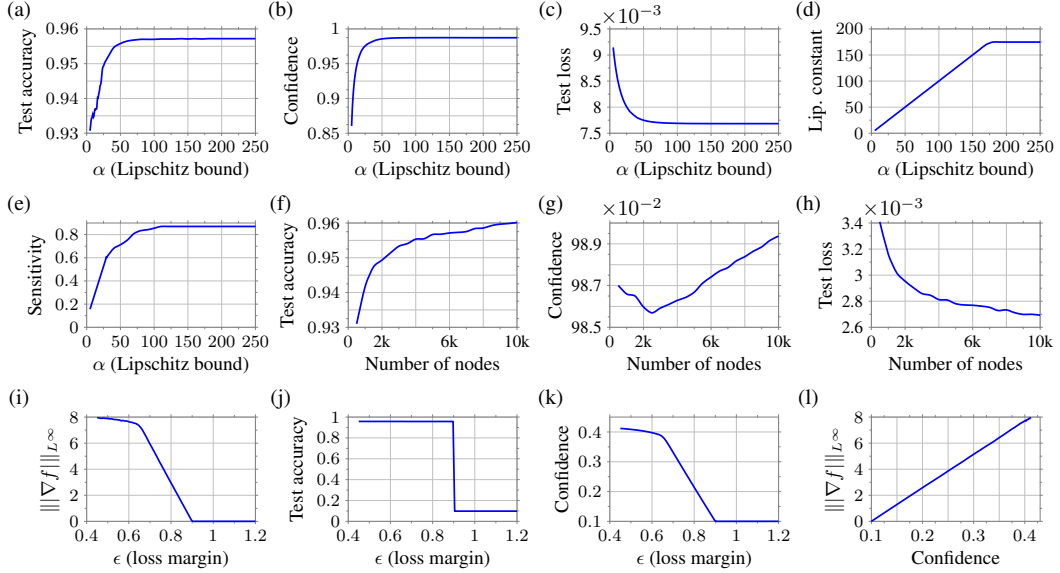

Figure 3: For the standard MNIST dataset, panels (a)-(c) show the relationships between Lipschitz bound, accuracy, confidence, and loss for Problem (2). (d) shows the relationship between the Lipschitz constant of the trained model and the Lipschitz bound in (2). We see that the Lipschitz constraint is active for $\alpha < 175$, and inactive otherwise. (e) shows the confidence degradation under bounded perturbation as we vary the Lipschitz bound in (2). Panels (f)-(h) show the dependence of accuracy, confidence and loss in testing on model complexity (number of vertices) for graph-based learning. Panels (i)-(k) show the dependence of the Lipschitz constant, accuracy and confidence in testing on the loss margin, for Problem (10). Panel (l) shows the tradeoff between performance and robustness, seen as an increase in the model Lipschitz constant with confidence in testing.

## 5   Conclusion

In this paper we propose a novel framework to train models with provable robustness guarantees. At its core, our framework relies on formulating a provably robust learning problem as a (convex) Lipschitz constrained loss minimization problem, for which we characterize and compute the solution by graph-based discretization and discrete heat flows. Our analysis defines a link between the properties of elliptic operators and adversarial learning, which provides us with a new perspective and powerful tools to investigate robustness properties of the minimizers. Following a similar analysis, we also study the complementary problem of improving the robustness of a model under a margin on the loss. We show that the two notions are tightly related, and that improving robustness necessarily leads to the deterioration of the performance of the model (in typical regimes). This robustification problem, which can be solved using an iterative procedure based on discrete heat flows involving the $p$-Laplacian, leads to the characterization of a fundamental tradeoff between the robustness of a model and its loss, thereby extending and generalizing recent results relating robustness and performance in adversarial machine learning. We illustrate our results via academic and a standard benchmark.

The ideas presented in this paper are of broad interest to the machine learning community and potentially open up a number of research directions. For instance, quantifying the optimality gap of minimizers of (7) with respect to the minimizer for Problem (2), for finite values of $n$ and $N$, under different Lipschitz bounds, interpolation schemes, and graph structures, will shed light on the underlying fundamental relationships between model complexity, generalization performance, and robustness in graph-based learning.

# 6    Broader impact

This paper is primarily of a theoretical nature. We expect our findings to impact the development of a formal theory of adversarially robust learning. Furthermore, we expect the proposed robust training schemes to contribute to efforts in adversarially robust graph-based learning. However, we do not envision any immediate application of our results to a societally relevant problem.

# 7    Funding disclosure

This work was supported in part by awards ARO-71603NSYIP, ONR-N00014-19-1-2264, and AFOSR-FA9550-20-1-0140.

## Footnotes

[1]We let $\mu$ be the underlying measure on $\mathbb{X}$, since the input data is generated from $\mu$ on the support $\mathbb{X}$.

[2]See Supplementary Material for a proof.

[3]We use $\nabla_1 \ell$ to denote the gradient of $\ell$ with respect to its first argument.

[4]See supplementary material for a proof.

[5]The constraint violation set is of zero measure, that is, $\mu(\{x \in \mathbb{X} \mid G_f(x) > 0\}) = 0$.

[6]Any $\lambda \in L^\infty(\mathbb{X}, \mu)^*$ is also a bounded, finitely additive (absolutely continuous) measure on $\mathbb{X}$.

[7]The $p$-Laplace operator is defined as $\Delta_p^\mu u = \frac{1}{\mu} \nabla \cdot \left( \mu |\nabla u|^{p-2} \nabla u \right)$.

[8]The code from numerical experiments in this paper is available on GitHub: `https://github.com/abedmakdah/Lipschitz-Bounds-and-Provably-Robust-Training-by-Laplacian-Smoothing.git`

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
