[Supplementary Material]

# Supplementary Material:
# Lipschitz Bounds and Provably Robust Training by Laplacian Smoothing

## A  Mathematical preliminaries

We introduce some mathematical preliminaries related to function spaces useful in developing our results. In what follows, we let $\mathbb{X} \subset \mathbb{R}^{\dim(\mathbb{X})}$ and $\mathbb{Y} \subset \mathbb{R}^{\dim(\mathbb{Y})}$ be compact and convex.

$L^p$ **and** $W^{1,p}$ **spaces.** The space $L^p(\mathbb{X}, \mu)$ of $p$-integrable functions on $\mathbb{X}$ with respect to an underlying (absolutely continuous) probability measure $\mu \in \mathcal{P}(\mathbb{X})$, is defined as:

$$L^p(\mathbb{X}, \mu) = \left\{ f : \mathbb{X} \to \mathbb{R} \;\middle|\; f \text{ measurable }, \; \int_{\mathbb{X}} |f|^p d\mu < \infty \right\}.$$

The Sobolev space $W^{1,p}(\mathbb{X}, \mu)$ is defined as:

$$W^{1,p}(\mathbb{X}, \mu) = \left\{ f \in L^p(\mathbb{X}, \mu) \;\middle|\; \int_{\mathbb{X}} |\nabla f|^p d\mu < \infty \right\}.$$

For $p = \infty$ in the above definitions, we get the space $L^\infty(\mathbb{X}, \mu)$ of essentially bounded measurable functions on $(\mathbb{X}, \mu)$ and the space $W^{1,\infty}(\mathbb{X}, \mu)$ of essentially bounded measurable functions with essentially bounded measurable gradients on $(\mathbb{X}, \mu)$.

Now, for $1 \leq p \leq \infty$, $L^p((\mathbb{X}, \mu); \mathbb{Y})$ is the space of measurable maps from $\mathbb{X}$ to $\mathbb{Y}$ such that $|f| \in L^p(\mathbb{X}, \mu)$ for any $f \in L^p((\mathbb{X}, \mu); \mathbb{Y})$, where $|\cdot|$ is the H-S norm in $\mathbb{Y}$. Moreover, $W^{1,p}((\mathbb{X}, \mu); \mathbb{Y})$ is the space of measurable maps such that $|f| \in L^p(\mathbb{X}, \mu)$ and $|\nabla f| \in L^p(\mathbb{X}, \mu)$ for any $f \in W^{1,p}((\mathbb{X}, \mu); \mathbb{Y})$.

**Lipschitz-continuous maps.** The space $\mathrm{Lip}(\mathbb{X}; \mathbb{Y})$ of Lipschitz-continuous maps from $\mathbb{X}$ to $\mathbb{Y}$ is such that for any $f \in \mathrm{Lip}(\mathbb{X}; \mathbb{Y})$, we have $|f(x_1) - f(x_2)| \leq \mathrm{lip}(f) |x_1 - x_2|$, where $\mathrm{lip}(f)$ is the Lipschitz constant of $f$. From Rademacher's theorem [1], every $f \in \mathrm{Lip}(\mathbb{X}; \mathbb{Y})$ is almost everywhere differentiable in $\mathbb{X}$ (with (a.e.) gradient $\nabla f$, which is also its weak gradient). Further, $\| |\nabla f| \|_{L^\infty(\mathbb{X})} = \mathrm{lip}(f)$ and we get $\mathrm{Lip}(\mathbb{X}; \mathbb{Y}) = W^{1,\infty}(\mathbb{X}; \mathbb{Y})$.

## B  Robustness to adversarial perturbations and the Lipschitz constant

In this section, we establish the dependence of sensitivity to adversarial perturbations of the loss on the Lipschitz constant of the input-output map. Recall from (1) that the loss $L_\sigma$ is given by:

$$L_\sigma(f) = \mathbb{E}_{(x,y) \sim \sigma} \left[ \ell(f(x), y) \right].$$

Adversarial perturbations [2] are a subset of perturbations on the data-generating distribution $\sigma$ generated by bounded maps $T$ that perturb the inputs $x \in \mathbb{X}$ while preserving the outputs $y \in \mathbb{Y}$. We illustrate this for a classification problem: Let $(x, y)$ be a true input-label pair in the (nominal) dataset and $f$ be a classifier that locally assigns to an input $x \in \mathbb{X}$ the label $f(x) \in \mathbb{Y}$. Let $r$ be a minimal perturbation on the input $x$, given a target label $y' \in \mathbb{Y}$, such that $f(x + r) = y'$ (where $y'$ is typically chosen to be an incorrect label for $x$, that is, $y' \neq y$). Now, an adversarial perturbation for

the classifier $f$ is generated by the replacement of $(x, y)$ by $(x + r, y)$ in the dataset. To formalize this, we define the class of maps:

$$\mathcal{T} = \{T \mid T(x, y) = (T_1(x, y) \, , \, y), \text{ and } T_1(x, y) \in B_\delta(x) \cap \mathbb{X}\},$$

where $B_\delta(x)$ is the open ball in $\mathbb{R}^{\dim(\mathbb{X})}$ of radius $\delta > 0$ and centered at $x$. Now, adversarial perturbations on the data-generating distribution $\sigma$ are a subset of perturbations generated by the class $\mathcal{T}$.

We first characterize the bound on the perturbation of the loss due to perturbations on $\sigma$ generated by the class $\mathcal{T}$. The perturbation by $T \in \mathcal{T}$ of the probability measure $\sigma$ yields the perturbed probability measure $T_\# \sigma$, where $T_\# \sigma$ is the pushforward of $\sigma$ by the map $T$[1]. We note that the perturbation of the loss $\left| L_{T_\# \sigma}(f) - L_\sigma(f) \right|$ satisfies:

$$
\begin{aligned}
\left| L_{T_\# \sigma}(f) - L_\sigma(f) \right| &= \left| \mathbb{E}_{(x,y) \sim T_\# \sigma}[\ell(f(x), y)] - \mathbb{E}_{(x,y) \sim \sigma}[\ell(f(x), y)] \right| \\
&= \left| \int_{\mathbb{X} \times \mathbb{Y}} \ell(f(x), y) d\,(T_\# \sigma)\,(x, y) - \int_{\mathbb{X} \times \mathbb{Y}} \ell(f(x), y) d\sigma(x, y) \right| \\
&= \left| \int_{\mathbb{X} \times \mathbb{Y}} \left( \ell(f(T_1(x, y)), y) - \ell(f(x), y) \right) d\sigma(x, y) \right| \\
&\le \operatorname{lip}(\ell)\operatorname{lip}(f) \left| \int_{\mathbb{X}} (T_1(x, y) - x)\, d\mu(x) \right| \\
&\le \operatorname{lip}(\ell)\operatorname{lip}(f)\delta.
\end{aligned}
$$

We next characterize the sensitivity of the loss for a given $f$ to perturbations on the data-generating distribution generated by the class $\mathcal{T}$. Let a family of transport maps $T^h = (1 - h)\operatorname{Id} + hT$ for some $T \in \mathcal{T}$ and $h \in [0, 1]$ (with Id being the identity map), perturb the data-generating distribution $\sigma$ as $\sigma^h = T_\#^h \sigma$. The (Gateaux) derivative of the loss along the family of adversarial perturbations $T^h$, is now given by:

$$
\begin{aligned}
D^{(T)} L_\sigma(f) = \left. \frac{d}{dh} L_{\sigma^h}(f) \right|_{h=0} &= \lim_{h \to 0} \frac{L_{\sigma^h}(f) - L_\sigma(f)}{h} \\
&= \lim_{h \to 0} \frac{1}{h} \int_{\mathbb{X} \times \mathbb{Y}} \left[ \ell(f(T^h(x, y)), y) - \ell(f(x), y) \right] d\sigma(x, y).
\end{aligned}
$$

We note that $\left| \frac{\ell(f(T^h(x,y)),y) - \ell(f(x),y)}{h} \right| \le \operatorname{lip}(\ell) \left| \frac{f(T^h(x,y)) - f(x)}{h} \right| \le \operatorname{lip}(\ell)\operatorname{lip}(f) \frac{\left| T^h(x,y) - x \right|}{h} = \operatorname{lip}(\ell)\operatorname{lip}(f) \left| T_1(x, y) - x \right|$. It then follows from the Dominated Convergence Theorem [3] that:

$$
\begin{aligned}
D^{(T)} L_\sigma(f) &= \int_{\mathbb{X} \times \mathbb{Y}} \langle \nabla_1 \ell(f(x), y) \cdot \nabla f(x) \, , \, T_1(x, y) - x \rangle \, d\sigma(x, y) \\
&= \mathbb{E}_{(x,y) \sim \sigma} \left[ \langle \nabla_1 \ell(f(x), y) \cdot \nabla f(x) \, , \, T_1(x, y) - x \rangle \right].
\end{aligned}
$$

We now define the sensitivity as the worst-case increase of the loss functional following an adversarial perturbation. That is, the sensitivity of the loss is the $L^\infty$-norm (with respect to the measure $\sigma$) of the gradient $\nabla_1 \ell \cdot \nabla f$ (precisely, $\|| \nabla_1 \ell \cdot \nabla f |\|_{L^\infty(\mathbb{X} \times \mathbb{Y}, \sigma)}$), which satisfies the bound:

$$
\underbrace{\|| \nabla_1 \ell \cdot \nabla f |\|_{L^\infty(\mathbb{X} \times \mathbb{Y}, \sigma)}}_{\text{sensitivity of } L \text{ to adv. perturbation}} \le \underbrace{\|| \nabla_1 \ell |\|_{L^\infty(\mathbb{X} \times \mathbb{Y}, \sigma)}}_{\text{Lipschitz constant of } \ell} \cdot \underbrace{\|| \nabla f |\|_{L^\infty(\mathbb{X}, \mu)}}_{\text{Lipschitz constant of } f}
$$

where $\mu$ is the marginal of $\sigma$ over $\mathbb{X}$, and $\|| \nabla f |\|_{L^\infty(\mathbb{X}, \mu)}$ is the Lipschitz constant of $f$ over the support of $\mu$.

We therefore get that the sensitivity of the loss functional to adversarial perturbations is indeed modulated by the Lipschitz constant of the input-output mapping. Thus, restricting the search space to the class of Lipschitz maps with a bound $\alpha \ge 0$ on the Lipschitz constant, as in the minimization problem (1), is convenient for analysis, and does not restrict the generality of the adversarially robust learning problem, and it allows us to obtain adversarially robust minimizers of the loss $L_\sigma$.

## C  The Lipschitz-constrained loss minimization problem (1) is convex

We recall that Problem (1) is given by:

$$\inf_{f \in \mathrm{Lip}(\mathbb{X},\mu)} \left\{ \underbrace{\mathbb{E}_{(x,y)\sim\sigma}\left[\ell\left(f(x),y\right)\right]}_{\triangleq L_\sigma(f)} \qquad \text{s.t. } \mathrm{lip}(f) \le \alpha \right\},$$

where $\sigma$ is an absolutely continuous probability measure on $\mathbb{X}\times\mathbb{Y}$ and the loss function $\ell : \mathbb{Y}\times\mathbb{Y} \to \mathbb{R}_{\ge 0}$ is strictly convex and Lipschitz continuous and $\alpha \ge 0$.

Firstly, we get that the loss $L_\sigma$ in (1) is strictly convex. To see this, let $f_1, f_2 \in \mathrm{Lip}(\mathbb{X},\mu)$ be such that $L_\sigma(f_1) < \infty$ and $L_\sigma(f_2) < \infty$. For $t \in [0,1]$, we get from the convexity of $\mathrm{Lip}(\mathbb{X},\mu)$ that $tf_1 + (1-t)f_2 \in \mathrm{Lip}(\mathbb{X},\mu)$. Also, from the strict convexity of the loss function $\ell$, we get:

$$
\begin{aligned}
L_\sigma(tf_1 + (1-t)f_2) &= \mathbb{E}_{(x,y)\sim\sigma}\left[\ell((tf_1+(1-t)f_2)(x),y)\right] \\
&= \mathbb{E}_{(x,y)\sim\sigma}\left[\ell(tf_1(x)+(1-t)f_2(x),y)\right] \\
&\le \mathbb{E}_{(x,y)\sim\sigma}\left[t\ell(f_1(x),y)+(1-t)\ell(f_2(x),y)\right] \\
&= t\mathbb{E}_{(x,y)\sim\sigma}\left[\ell(f_1(x),y)\right]+(1-t)\mathbb{E}_{(x,y)\sim\sigma}\left[\ell(f_2(x),y)\right] \\
&= tL_\sigma(f_1)+(1-t)L_\sigma(f_2).
\end{aligned}
$$

Moreover, the inequality is strict for $t \in (0,1)$, from which it follows that the loss $L_\sigma$ is strictly convex.

Now, let $f_1, f_2 \in \mathrm{Lip}(\mathbb{X},\mu)$ such that $\mathrm{lip}(f_1) \le \alpha$ and $\mathrm{lip}(f_2) \le \alpha$. For the map $\lambda f_1 + (1-\lambda)f_2$, $\lambda \in [0,1]$, and $x_1, x_2 \in \mathbb{X}$, it follows that:

$$
\begin{aligned}
|(\lambda f_1 + (1-\lambda)f_2)(x_1) &- (\lambda f_1 + (1-\lambda)f_2)(x_2)| \\
&= |\lambda\left(f_1(x_1)-f_1(x_2)\right)+(1-\lambda)(f_2(x_1)-f_2(x_2))| \\
&\le \lambda|f_1(x_1)-f_1(x_2)|+(1-\lambda)|f_2(x_1)-f_2(x_2)| \\
&\le \lambda\mathrm{lip}(f_1)|x_1-x_2|+(1-\lambda)\mathrm{lip}(f_2)|x_1-x_2| \\
&\le \alpha|x_1-x_2|,
\end{aligned}
$$

and we get $\mathrm{lip}(\lambda f_1+(1-\lambda)f_2) \le \alpha$. Therefore, the constraint in (1) is convex. From strict convexity of the loss $L_\sigma$ and convexity of the constraint set $\{f \in \mathrm{Lip}((\mathbb{X},\mu),\mathbb{Y}) \mid \mathrm{lip}(f) \le \alpha\}$, we get that Problem (1) is convex.

## D  Proof of Theorem 2.1 (Saddle point of Lagrangian $\mathcal{L}$)

*(i) Derivative of loss function $L$ w.r.t $f$.* We have:

$$L_\sigma(f) = \mathbb{E}_{x\sim\mu}\left[\mathbb{E}_{y\sim\pi(y\mid x)}\left[\ell(f(x),y)\right]\right],$$

where $\mu$ is the marginal over $\mathbb{X}$ and $\pi$ the conditional of the joint distribution $\sigma \in \mathcal{P}(\mathbb{X}\times\mathbb{Y})$. Let $\{f^\epsilon\}_{\epsilon\in[0,1]}$ be a family of maps from $\mathbb{X}$ to $\mathbb{Y}$ that is pointwise smooth (i.e., for any $x \in \mathbb{X}$, $F(\epsilon,x) = f^\epsilon(x)$ is smooth in $\epsilon$). We now evaluate the derivative of the loss function $L_\sigma$ w.r.t. the family $\{f^\epsilon\}_{\epsilon\in[0,1]}$, at $\epsilon = 0$, as follows:

$$
\begin{aligned}
\frac{dL_\sigma}{d\epsilon}(f^0) &= \lim_{\epsilon\to 0}\frac{L_\sigma(f^\epsilon)-L_\sigma(f^0)}{\epsilon} \\
&= \lim_{\epsilon\to 0}\frac{1}{\epsilon}\int_{\mathbb{X}}\left[\int_{\mathbb{Y}}\left(\ell(f^\epsilon(x),y)-\ell(f^0(x),y)\right)d\pi(y\mid x)\right]d\mu(x).
\end{aligned}
$$

We note that $\left|\frac{\ell(f^\epsilon(x),y)-\ell(f^0(x),y)}{\epsilon}\right| \le \mathrm{lip}(\ell)\left|\frac{f^\epsilon(x)-f^0(x)}{\epsilon}\right| \le \mathrm{lip}(\ell)\mathrm{lip}(F(\cdot,x))$, where $\mathrm{lip}(F(\cdot,x))$ is the Lipschitz constant of $F$ as a function of $\epsilon$ at every $x \in \mathbb{X}$ (since $F(\cdot,x)$ is smooth in $[0,1]$ for every $x \in \mathbb{X}$, it is also Lipschitz continuous). It then follows from the Dominated Convergence

Theorem [3] that:

$$\frac{dL_\sigma}{d\epsilon}(f^0) = \lim_{\epsilon \to 0} \frac{1}{\epsilon} \int_\mathbb{X} \left[ \int_\mathbb{Y} \left( \ell(f^\epsilon(x), y) - \ell(f^0(x), y) \right) d\pi(y \mid x) \right] d\mu(x)$$

$$= \int_\mathbb{X} \left[ \int_\mathbb{Y} \lim_{\epsilon \to 0} \frac{1}{\epsilon} \left( \ell(f^\epsilon(x), y) - \ell(f^0(x), y) \right) d\pi(y \mid x) \right] d\mu(x)$$

$$= \int_\mathbb{X} \left[ \int_\mathbb{Y} \nabla_1 \ell(f^0(x), y) \cdot \left. \frac{\partial f^\epsilon}{\partial \epsilon}(x) \right|_{\epsilon=0} d\pi(y \mid x) \right] d\mu(x)$$

$$= \int_\mathbb{X} \left[ \int_\mathbb{Y} \nabla_1 \ell(f^0(x), y) \, d\pi(y \mid x) \right] \cdot \left. \frac{\partial f^\epsilon}{\partial \epsilon}(x) \right|_{\epsilon=0} d\mu(x)$$

$$= \int_\mathbb{X} \frac{\partial \bar{L}}{\partial f} \cdot \left. \frac{\partial f^\epsilon}{\partial \epsilon} \right|_{\epsilon=0} d\mu(x),$$

where we denote by $\partial_f \bar{L}_\sigma = \frac{\partial \bar{L}_\sigma}{\partial f} = \int_\mathbb{Y} \nabla_1 \ell(f^0(x), y) \, d\pi(y \mid x)$ the functional derivative of $\bar{L}_\sigma$ w.r.t. $f$.

*(ii) Minimizer of* (2). The search space for Problem (2) is given by,

$$\mathcal{F} = \left\{ f \in W^{1,\infty}((\mathbb{X}, \mu), \mathbb{Y}) \mid \||\nabla f|\|_{L^\infty(\mathbb{X}, \mu)} \leq \alpha \right\}.$$

We see that $\mathcal{F}$ is closed, convex and bounded. Boundedness of $\mathcal{F}$ follows from compactness of $\mathbb{Y}$ which implies that there exists an $M \in \mathbb{R}_{\geq 0}$ such that $\mathbb{Y} \subset B_M(\mathbf{0}_\mathbb{Y})$. It follows that for any $f \in \mathcal{F}$, we have $\||f|\|_{L^\infty(\mathbb{X}, \mu)} \leq M$. Moreover, we have $\||\nabla f|\|_{L^\infty(\mathbb{X}, \mu)} \leq \alpha$. Therefore, $\|f\|_{W^{1,\infty}((\mathbb{X}, \mu), \mathbb{Y})} = \||f|\|_{L^\infty(\mathbb{X}, \mu)} + \||\nabla f|\|_{L^\infty(\mathbb{X}, \mu)} \leq M + \alpha < \infty$ for any $f \in \mathcal{F}$.

The loss $L_\sigma$ is strictly convex and lower semicontinuous (in fact, it is (Gateaux) differentiable as seen earlier for absolutely continuous $\sigma$, since $\ell$ is strictly convex and Lipschitz-continuous).

Let $\{f_n\}_{n \in \mathbb{N}}$ be a minimizing sequence in $\mathcal{F}$ for the loss $L_\sigma$, such that $f_n \in \mathcal{F}$ and $\lim_{n \to \infty} L_\sigma(f_n) = \inf_{f \in \mathcal{F}} L_\sigma(f)$. Clearly, the sequence $\{f_n\}_{n \in \mathbb{N}}$ is uniformly bounded since $\|f_n\|_{W^{1,\infty}((\mathbb{X}, \mu), \mathbb{Y})} \leq M + \alpha$. It is also uniformly equicontinuous, since $|f_n(x_1) - f_n(x_2)| \leq \alpha |x_1 - x_2|$ for all $n \in \mathbb{N}$. Therefore, by the Arzelà-Ascoli Theorem [3], there exists a uniformly converging subsequence $\{f_{n_j}\}_{j \in \mathbb{N}}$, with the limit $f^* \in \mathcal{F}$. Furthermore, by the continuity of $L_\sigma$, we get $\lim_{j \to \infty} L_\sigma(f_{n_j}) = L_\sigma(f^*) = \min_{f \in \mathcal{F}} L_\sigma(f)$. By the strict convexity of the loss $L_\sigma$, we get that $f^*$ is the unique global minimizer of $L_\sigma$.

Thus, Problem (2) has a unique global minimizer $f^* \in \left\{ f \in W^{1,\infty}((\mathbb{X}, \mu), \mathbb{Y}) \mid \text{lip}(f) \leq \alpha \right\}$.

*(iii) Saddle points of Lagrangian functional* $\mathcal{L}_\sigma$. The constraint set is given by $\{f \in W^{1,\infty}((\mathbb{X}, \mu), \mathbb{Y}) \mid \mathcal{G}(f) \in (-\infty, 0]\}$, where $\mathcal{G}(f) = \|G_f\|_{L^\infty(\mathbb{X}, \mu)}$, and we have the constraint qualification:

$$0 \in \text{int} \left\{ \mathcal{G}\left( W^{1,\infty}((\mathbb{X}, \mu), \mathbb{Y}) \right) + [0, \infty) \right\},$$

where the operation $+$ denotes the Minkowski sum. This allows us to apply Theorem 3.6 in [4] to infer that the set of Lagrange multipliers corresponding to the (unique) minimizer $f^*$ is a non-empty, convex, bounded and weakly$-^*$ compact subset of $L^\infty(\mathbb{X}, \mu)^*_{\geq 0}$. Moreover, we note that $(-\infty, 0]$ is a closed convex cone, and it follows from Theorem 3.4-(iii) in [4] that for any Lagrange multiplier $\lambda^*$, the pair $(f^*, \lambda^*)$ is a saddle point of the Lagrangian functional $\mathcal{L}_\sigma$. Uniqueness of $\lambda^*$ again follows from the strict convexity of $L_\sigma$. We also have the *feasibility* condition $G_{f^*} \leq 0$ (that is, $|\nabla f^*| \leq \alpha$) and $\lambda^* \geq 0$ $\mu$-a.e. in $\mathbb{X}$.

Now, the (Gateaux) derivative of the Lagrangian $\mathcal{L}_\sigma(f, \lambda) = L_\sigma(f) + \lambda(G_f)$ in $W^{1,\infty}((\mathbb{X}, \mu), \mathbb{Y})$ along $V \in W^{1,\infty}((\mathbb{X}, \mu), \mathbb{Y})$ is given by:

$$D_1^{(V)} \mathcal{L}_\sigma(f, \lambda) = \int_\mathbb{X} \partial_f \bar{L}_\sigma \cdot V \, d\mu + \int_\mathbb{X} \nabla f \cdot \nabla V \, d(\lambda \mu),$$

where $D_1^{(V)}$ denotes the directional derivative of the first argument along $V$ and $\lambda \mu$ is an absolutely continuous measure ($\lambda$-weighting on the underlying measure $\mu$. Recall that $\lambda \in L^\infty(\mathbb{X}, \mu)^*_{\geq 0}$ is itself a bounded, finitely additive absolutely continuous measure). The above expression can be derived

using a similar construction of a limit and the application of the Dominated Convergence Theorem as earlier in this section.

By the Minimax Theorem, we have $\mathcal{L}_\sigma(f^*, \lambda^*) = \inf_f \sup_\lambda \mathcal{L}_\sigma(f, \lambda) = \sup_\lambda \inf_f \mathcal{L}_\sigma(f, \lambda)$, where the infimum is taken over $W^{1,\infty}((\mathbb{X}, \mu), \mathbb{Y})$ and the supremum over $\lambda \in L^\infty(\mathbb{X}, \mu)^*_{\geq 0}$. We therefore have $\mathcal{L}_\sigma(f^*, \lambda^*) \geq \mathcal{L}_\sigma(f^*, 0)$, which yields the condition $\lambda^*(G_{f^*}) \geq 0$. Moreover, from feasibility, we have $G_{f^*} \leq 0$ and $\lambda^* \geq 0$, which implies that $\lambda^*(G_{f^*}) \leq 0$. This results in the *complementary slackness* condition $\lambda^*(G_{f^*}) = 0$. From the Minimax equality, we get that $(f^*, \lambda^*)$ is also a critical point of $\mathcal{L}_\sigma$, that is, $D_1^{(V)}\mathcal{L}_\sigma(f^*, \lambda^*) = 0$, which implies that $\int_{\mathbb{X}} \partial_f \bar{L}_\sigma(f^*) \cdot V \, d\mu + \int_{\mathbb{X}} \nabla f^* \cdot \nabla V \, d(\lambda^*\mu) = 0$, which is the *stationarity* condition.

*(iv) Improved regularity of Lagrange multiplier $\lambda^*$.* We can indeed establish stronger regularity for the Lagrange multiplier $\lambda^*$. We have that the Lagrange multipliers $\lambda^* \in L^\infty(\mathbb{X}, \mu)^*_{\geq 0}$, which is a bounded, finitely additive measure absolutely continuous measure, is also a linear continuous functional on $L^\infty(\mathbb{X}, \mu)$ and must therefore vanish on sets of $\mu$-measure zero (i.e., $\lambda^*(A) = 0$ for $A \subset \mathbb{X}$ with $\mu(A) = 0$). Moreover, from Theorem 1.24 in [5], we can decompose $\lambda^* = \lambda^*_c + \lambda^*_p$, where $\lambda^*_c$ is a non-negative countably additive measure and $\lambda^*_p$ is non-negative and purely finitely additive. By the Radon-Nikodym theorem, we get that there exists a function $h_c \in L^1(\mathbb{X}, \mu)$ such that the countably additive and absolutely continuous measure $\lambda^*_c$ satisfies $d\lambda^*_c = h_c \, d\mu$. By substitution in the stationarity condition, we get $\int_{\mathbb{X}} \partial_f \bar{L}_\sigma \cdot V d\mu = -\int_{\mathbb{X}} \nabla f^* \cdot \nabla V \, d(\lambda^*_c\mu) - \int_{\mathbb{X}} \nabla f^* \cdot \nabla V \, d(\lambda^*_p\mu)$. We now consider a set $D_\delta = \{x \in \mathbb{X} \mid -\delta \leq G_{f^*}(x) \leq 0\}$, with $0 < \delta < \alpha^2$. By complementary slackness, we note that $\lambda^*(\mathbb{X} \setminus D_\delta) = 0$. Since $\lambda^*_p$ is purely finitely additive, it implies that there must exist a collection of nonempty sets $\{E_n\}_{n\in\mathbb{N}}$ with $E_{n+1} \subset E_n$ and $\lim_{n\to\infty} E_n = \emptyset$, such that $\lim_{n\to\infty} \lambda^*_p(E_n) > 0$[2]. Since $\lambda^*(\mathbb{X} \setminus D_\delta) = 0$, we can suppose without loss of generality that $E_0 \subset D_\delta$. We also consider another collection of nonempty sets $\{E'_n\}_{n\in\mathbb{N}}$, with the same properties (with $E'_0 \subset D_\delta$, $E'_{n+1} \subset E'_n$ and $\lim_{n\to\infty} E'_n = \emptyset$), such that $E_n \subset E'_n$ for all $n \in \mathbb{N}$. We note that for $x \in D_\delta$, we have $0 < \alpha^2 - \delta \leq |\nabla f^*(x)|^2 \leq \alpha^2$, which implies that $\nabla f^*$ does not vanish on $E'_n$ for any $n \in \mathbb{N}$. We now consider a family of variations $V_n \in W^{1,\infty}(\mathbb{X}, \mu)$ for $n \in \mathbb{N}$ such that $V_n$ and $\nabla V_n$ are supported in $E'_n$, $\nabla f^* \cdot \nabla V_n \geq 0$ in $E'_n$ and $\nabla f^* \cdot \nabla V_n \geq \epsilon$ in $E_n$ (uniformly). The stationarity condition now yields, for $n \in \mathbb{N}$:

$$-\int_{E'_n} \partial_f \bar{L}_\sigma(f^*) \cdot V_n d\mu = \int_{E'_n} (\nabla f^* \cdot \nabla V_n) \, h_c d\mu + \int_{E'_n} \nabla f^* \cdot \nabla V_n \, d(\lambda^*_p\mu)$$
$$\geq \int_{E'_n} (\nabla f^* \cdot \nabla V_n) \, h_c d\mu + \epsilon \int_{E_n} d(\lambda^*_p\mu).$$

In the limit $n \to 0$, we have $\lim_{n\to\infty} \int_{E'_n} \partial_f \bar{L}_\sigma(f^*) \cdot V_n d\mu = 0$ and $\lim_{n\to\infty} \int_{E'_n} (\nabla f^* \cdot \nabla V_n) \, h_c d\mu = 0$, which implies that $0 \leq \lim_{n\to\infty} \epsilon \int_{E_n} d(\lambda^*_p\mu) \leq 0$, and we get $\lim_{n\to\infty} \lambda^*_p(E_n) = 0$, i.e., the measure $\lambda^*$ does not have a purely finitely additive component. Therefore, the measure $\lambda^*$ is countably additive (and absolutely continuous) and possesses a Radon-Nikodym derivative w.r.t. $\mu$, in $L^1(\mathbb{X}, \mu)$. For ease of notation, we henceforth let $\lambda^* \in L^1(\mathbb{X}, \mu)$ also denote its density function.

Since $\lambda^* \in L^1(\mathbb{X}, \mu)_{\geq 0}$ and $G_{f^*} \leq 0$ $\mu$-a.e. in $\mathbb{X}$, we can now indeed state the complementary slackness condition as $\lambda^* (|\nabla f^*| - \alpha) = 0$ $\mu$-a.e. in $\mathbb{X}$.

Moreover, the stationarity condition, under $\lambda^* \in L^1(\mathbb{X}, \mu)_{\geq 0}$ can now be expressed as:

$$0 = \int_{\mathbb{X}} \partial_f \bar{L}_\sigma(f^*) \cdot V \, d\mu + \int_{\mathbb{X}} \nabla f^* \cdot \nabla V \, \lambda^* \, d\mu$$
$$= \int_{\mathbb{X}} \partial_f \bar{L}_\sigma(f^*) \cdot V \, d\mu - \int_{\mathbb{X}} \frac{1}{\mu} \nabla \cdot (\lambda^* \mu \nabla f^*) \cdot V \, d\mu + \int_{\partial\mathbb{X}} \lambda^* \nabla f^* \cdot \mathbf{n} V \mu \, dS,$$

where we have used the Divergence Theorem to obtain the final equality, with $S$ as the surface measure on $\partial\mathbb{X}$. As the above holds for any variation $V \in W^{1,\infty}((\mathbb{X}, \mu), \mathbb{Y})$, it must follow that $-\frac{1}{\mu}\nabla \cdot (\mu\lambda^*\nabla f^*) + \partial_f \bar{L}_\sigma(f^*) = 0$ $\mu$-a.e. in $\mathbb{X}$ and $\lambda^*\mu\nabla f^* \cdot \mathbf{n} = 0$ on $\partial\mathbb{X}$, and if we do not suppose stronger regularity of the saddle point $(f^*, \lambda^*)$, the equations must be hold weakly.

The above correspond to the necessary KKT conditions. Conversely, any solution pair $(f^*, \lambda^*)$ which satisfies the above KKT conditions is a saddle point for the Lagrangian $\mathcal{L}_\sigma$ and is a solution to the original optimization problem.

## E    Proof of Theorem 3.1 (Saddle points of Lagrangian $\mathcal{H}$)

*(i) Minimizers of* (11). The search space for Problem (11) is given by:

$$\mathcal{F}_p = \left\{ f \in W^{1,p}((\mathbb{X}, \mu), \mathbb{Y}) \mid L_\sigma(f) \le J_\sigma^*(\alpha) + \epsilon \right\}.$$

Let $\{u_n\}_{n \in \mathbb{N}}$ be a minimizing sequence in $\mathcal{F}_p$ for the $W^{1,p}$-seminorm, such that $u_n \in \mathcal{F}_p$ for all $n \in \mathbb{N}$ and $\lim_{n \to \infty} \||\nabla u_n|\|_{L^p(\mathbb{X}, \mu)} = \inf_{u \in \mathcal{F}_p} \||\nabla u|\|_{L^p(\mathbb{X}, \mu)}$. Since $f^* \in W^{1,\infty}((\mathbb{X}, \mu), \mathbb{Y})$, the minimizer of Problem (2) also belongs to $\mathcal{F}_p$, that is, $f^* \in \mathcal{F}_p$ and $inf_{u \in \mathcal{F}_p} \||\nabla u|\|_{L^p(\mathbb{X}, \mu)} \le \||\nabla f^*|\|_{L^p(\mathbb{X}, \mu)} \le \alpha$, we can choose the minimizing sequence to satisfy the bound $\||\nabla u_n|\|_{L^p(\mathbb{X}, \mu)} \le \alpha$. Similar to Section D, we now have the uniform bound $\|u_n\|_{W^{1,p}((\mathbb{X}, \mu), \mathbb{Y})} \le M + \alpha$ for all $n \in \mathbb{N}$. For $p > \dim(\mathbb{X})$, we have from Morrey's Inequality [1], for every $n \in \mathbb{N}$, that:

$$|u_n(x_1) - u_n(x_2)| \le \frac{2p \dim(\mathbb{X})}{p - \dim(\mathbb{X})} |x_1 - x_2|^{1 - \frac{\dim(\mathbb{X})}{p}} \||\nabla u_n|\|_{L^p(\mathbb{X}, \mu)}$$
$$\le 2C \dim(\mathbb{X})(1 + \dim(\mathbb{X})) |x_1 - x_2|^{\frac{1}{1 + \dim(\mathbb{X})}} \alpha,$$

where $C = \max\left\{ 1, \operatorname{diam}(\mathbb{X})^{\frac{\dim(\mathbb{X})}{1 + \dim(\mathbb{X})}} \right\}$. Thus, the sequence $\{u_n\}_{n \in \mathbb{N}}$ is also uniformly equicontinuous. Therefore, by the Arzelà-Ascoli Theorem, there exists a uniformly converging subsequence $\{u_{n_j}\}_{j \in \mathbb{N}}$ with limit $f^{\epsilon, p} \in \mathcal{F}_p$. Furthermore, by the continuity of the $W^{1,p}$-seminorm, we get that $\lim_{j \to \infty} \||\nabla u_{n_j}|\|_{L^p(\mathbb{X}, \mu)} = \||\nabla f^{\epsilon, p}|\|_{L^p(\mathbb{X}, \mu)} = \min_{f \in \mathcal{F}_p} \||\nabla f|\|_{L^p(\mathbb{X}, \mu)}$. By convexity of the $W^{1,p}$-seminorm, we get that $f^{\epsilon, p}$ is a global minimizer for Problem (11).

We therefore conclude that Problem (11) is guaranteed to have (atleast one) global minimizer $f^{\epsilon, p} \in \left\{ f \in W^{1,p}((\mathbb{X}, \mu), \mathbb{Y}) \mid L_\sigma(f) \le J_\sigma^*(\alpha) + \epsilon \right\}$.

*(ii) Saddle points of Lagrangian functional* $\mathcal{H}_\sigma$. The constraint set is given by $\{f \in W^{1,p}((\mathbb{X}, \mu), \mathbb{Y}) \mid \mathcal{G}(f) \le 0\}$, where $\mathcal{G}(f) = L_\sigma(f) - (J_\sigma^*(\alpha) + \epsilon)$, and we have the constraint qualification:

$$0 \in \operatorname{int} \left\{ \mathcal{G}\left( W^{1,p}((\mathbb{X}, \mu), \mathbb{Y}) \right) + [0, \infty) \right\},$$

where the operation $+$ denotes the Minkowski sum. This allows us to apply Theorem 3.6 in [4] to infer that the set of Lagrange multipliers corresponding to the minimizer $f^{\epsilon, p}$ is a non-empty, convex, bounded and weakly$-^*$ compact subset of $\mathbb{R}_{\ge 0}$. Moreover, we note that $(-\infty, 0]$ is a closed convex cone, and it follows from Theorem 3.4-(iii) in [4] that for any Lagrange multiplier $\kappa^{\epsilon, p}$, the pair $(f^{\epsilon, p}, \kappa^{\epsilon, p})$ is a saddle point of the Lagrangian functional $\mathcal{H}_\sigma$. We also have the *feasibility* condition $L_\sigma(f) \le J_\sigma^*(\alpha) + \epsilon$.

Following a similar procedure as in Section D, we obtain the (Gateaux) derivative of the Lagrangian $\mathcal{H}_\sigma(f, \kappa) = \frac{1}{p} \||\nabla f|\|_{L^p(\mathbb{X}, \mu)}^p + \kappa \left( L_\sigma(f) - (J_\sigma^*(\alpha) + \epsilon) \right)$ in $W^{1,p}((\mathbb{X}, \mu), \mathbb{Y})$ along $V \in W^{1,p}((\mathbb{X}, \mu), \mathbb{Y})$ as:

$$D_1^{(V)} \mathcal{H}_\sigma(f, \kappa) = \int_{\mathbb{X}} |\nabla f|^{p-2} \nabla f \cdot \nabla V \, d\mu + \kappa \int_{\mathbb{X}} \partial_f \bar{L}_\sigma(f) \cdot V \, d\mu.$$

By the Minimax Theorem, we have $\mathcal{H}_\sigma(f^{\epsilon, p}, \kappa^{\epsilon, p}) = \inf_f \sup_\kappa \mathcal{H}_\sigma(f, \kappa) = \sup_\kappa \inf_f \mathcal{H}_\sigma(f, \kappa)$, where the infimum is taken over $W^{1,p}((\mathbb{X}, \mu), \mathbb{Y})$ and the supremum over $\mathbb{R}_{\ge 0}$. We therefore have $\mathcal{H}_\sigma(f^{\epsilon, p}, \kappa^{\epsilon, p}) \ge \mathcal{H}_\sigma(f^{\epsilon, p}, 0)$, which yields the condition $\kappa^{\epsilon, p} \left( L_\sigma(f^{\epsilon, p}) - (J_\sigma^*(\alpha) + \epsilon) \right) \ge 0$. Moreover, from feasibility, we have $L_\sigma(f^{\epsilon, p}) \le J_\sigma^*(\alpha) + \epsilon$ and $\kappa^{\epsilon, p} \ge 0$, which implies that $\kappa^{\epsilon, p} \left( L_\sigma(f^{\epsilon, p}) - (J_\sigma^*(\alpha) + \epsilon) \right) \le 0$. This results in the *complementary slackness* condition $\kappa^{\epsilon, p} \left( L_\sigma(f^{\epsilon, p}) - (J_\sigma^*(\alpha) + \epsilon) \right) = 0$. From the Minimax equality, we get that $(f^{\epsilon, p}, \kappa^{\epsilon, p})$ is

also a critical point of $\mathcal{H}_\sigma$, that is $D_1^{(V)}\mathcal{H}_\sigma(f^{\epsilon,p},\kappa^{\epsilon,p})=0$ for any $V \in W^{1,p}((\mathbb{X},\mu),\mathbb{Y})$:

$$
\begin{aligned}
0 &= \int_{\mathbb{X}} |\nabla f^{\epsilon,p}|^{p-2}\nabla f^{\epsilon,p} \cdot \nabla V \, d\mu + \kappa^{\epsilon,p} \int_{\mathbb{X}} \partial_f \bar{L}_\sigma(f^{\epsilon,p}) \cdot V \, d\mu \\
&= -\int_{\mathbb{X}} \frac{1}{\mu}\nabla \cdot \left(\mu|\nabla f^{\epsilon,p}|^{p-2}\nabla f^{\epsilon,p}\right) \cdot V \, d\mu + \int_{\partial\mathbb{X}} |\nabla f^{\epsilon,p}|^{p-2}\nabla f^{\epsilon,p} \cdot \mathbf{n}V\mu \, dS \\
&\quad + \kappa^{\epsilon,p} \int_{\mathbb{X}} \partial_f \bar{L}_\sigma(f^{\epsilon,p}) \cdot V \, d\mu,
\end{aligned}
$$

where we have used the Divergence Theorem to obtain the final equality, with $S$ as the surface measure on $\partial\mathbb{X}$. This is the *stationarity* condition. As the above holds for any variation $V \in W^{1,p}((\mathbb{X},\mu);\mathbb{Y})$, it must follow that $-\frac{1}{\mu}\nabla\cdot\left(\mu|\nabla f^{\epsilon,p}|^{p-2}\nabla f^{\epsilon,p}\right)+\kappa^{\epsilon,p}\partial_f\bar{L}_\sigma(f^{\epsilon,p})=0$ $\mu$-a.e. in $\mathbb{X}$ and $\mu\nabla f^{\epsilon,p}\cdot\mathbf{n}=0$ on $\partial\mathbb{X}$, and if we do not suppose stronger regularity of $f^{\epsilon,p}$, the equations must be hold weakly.

The above correspond to the necessary KKT conditions. Conversely, any solution pair $(f^{\epsilon,p},\kappa^{\epsilon,p})$ which satisfies the above KKT conditions is a saddle point for the Lagrangian $\mathcal{H}_\sigma$ and is a solution to the original optimization problem.

# F    Proof of Theorem 3.2 (Convergence as $p \to \infty$)

*(i) Monotonicity properties of $W^{1,p}((\mathbb{X},\mu);\mathbb{Y})$.* We first note that for $p,q \in \mathbb{N}$, $1 < p < q$ and an $f \in W^{1,p}((\mathbb{X},\mu);\mathbb{Y})$, $\|\|f\|\|_{L^p(\mathbb{X},\mu)} \leq \|\|f\|\|_{L^q(\mathbb{X},\mu)}$ and $\|\|\nabla f\|\|_{L^p(\mathbb{X},\mu)} \leq \|\|\nabla f\|\|_{L^q(\mathbb{X},\mu)}$. It follows that $W^{1,q}((\mathbb{X},\mu);\mathbb{Y}) \subseteq W^{1,p}((\mathbb{X},\mu);\mathbb{Y})$. In particular, for any $p \in \mathbb{N}$, $p > 1$, we have $\|\|f\|\|_{L^p(\mathbb{X},\mu)} \leq \|\|f\|\|_{L^\infty(\mathbb{X},\mu)}$, $\|\|\nabla f\|\|_{L^p(\mathbb{X},\mu)} \leq \|\|\nabla f\|\|_{L^\infty(\mathbb{X},\mu)}$ and $W^{1,\infty}((\mathbb{X},\mu);\mathbb{Y}) \subseteq W^{1,p}((\mathbb{X},\mu);\mathbb{Y})$. It then follows that $\left\{f \in W^{1,q}((\mathbb{X},\mu);\mathbb{Y}) \mid L_\sigma(f) \leq \epsilon\right\} \subseteq \left\{f \in W^{1,p}((\mathbb{X},\mu);\mathbb{Y}) \mid L_\sigma(f) \leq \epsilon\right\}$ for $1 < p < q \leq \infty$.

*(ii) Minimizers.* From the strict convexity of $L_\sigma$, it follows that $\left\{f \in W^{1,p}((\mathbb{X},\mu);\mathbb{Y}) \mid L_\sigma(f) \leq \epsilon\right\}$ is closed and convex for any $1 < p \leq \infty$. Moreover, the semi-norm of $f \in W^{1,p}((\mathbb{X},\mu);\mathbb{Y})$, i.e., $\|\|\nabla f\|\|_{L^p(\mathbb{X},\mu)}$, is convex. The existence of global minimizers for the problem:

$$
\inf_{f\in W^{1,p}((\mathbb{X},\mu);\mathbb{Y})} \left\{ \|\|\nabla f\|\|_{L^p(\mathbb{X},\mu)}, \qquad \text{s.t. } L_\sigma(f) \leq J_\sigma^*(\alpha) + \epsilon \right\}
$$

was established in Section E for every $\dim(\mathbb{X}) < p \leq \infty$ and $\epsilon > 0$.

*(iii) Monotonicity of minimum value.* From the existence of a global minimum value for any $\dim(\mathbb{X}) < p < \infty$, and the monotonicity properties of $W^{1,p}(\mathbb{X},\mu)$, we get for $\dim(\mathbb{X}) < p \leq q$:

$$
\min_{\substack{f\in W^{1,p}((\mathbb{X},\mu);\mathbb{Y}) \\ L_\sigma(f)\leq J_\sigma^*(\alpha)+\epsilon}} \|\|\nabla f\|\|_{L^p(\mathbb{X},\mu)} \leq \min_{\substack{f\in W^{1,q}((\mathbb{X},\mu);\mathbb{Y}) \\ L_\sigma(f)\leq J_\sigma^*(\alpha)+\epsilon}} \|\|\nabla f\|\|_{L^q(\mathbb{X},\mu)}.
$$

In particular, we get for any $p > \dim(\mathbb{X})$:

$$
\min_{\substack{f\in W^{1,p}((\mathbb{X},\mu);\mathbb{Y}) \\ L_\sigma(f)\leq J_\sigma^*(\alpha)+\epsilon}} \|\|\nabla f\|\|_{L^p(\mathbb{X},\mu)} \leq \min_{\substack{f\in \text{Lip}((\mathbb{X},\mu);\mathbb{Y}) \\ L_\sigma(f)\leq J_\sigma^*(\alpha)+\epsilon}} \text{lip}(f).
$$

Therefore, by the convergence of bounded monotone sequences, we get:

$$
\lim_{p\to\infty} \min_{\substack{f\in W^{1,p}((\mathbb{X},\mu);\mathbb{Y}) \\ L_\sigma(f)\leq J_\sigma^*(\alpha)+\epsilon}} \|\|\nabla f\|\|_{L^p(\mathbb{X},\mu)} \leq \min_{\substack{f\in \text{Lip}((\mathbb{X},\mu);\mathbb{Y}) \\ L_\sigma(f)\leq J_\sigma^*(\alpha)+\epsilon}} \text{lip}(f) = \bar{\alpha}(\epsilon).
$$

*(iv) Upper bound is indeed the supremum.* We now consider the sequence of minimizers $\{f_\sigma^{\epsilon,p}\}_{p\in\mathbb{N}}$:

$$
f_\sigma^{\epsilon,p} \in \arg\min_{\substack{f\in W^{1,p}((\mathbb{X},\mu);\mathbb{Y}) \\ L_\sigma(f)\leq J_\sigma^*(\alpha)+\epsilon}} \|\|\nabla f\|\|_{L^p(\mathbb{X},\mu)}.
$$

Fixing a $p > \dim(\mathbb{X})$, from the monotonicity of minimum values and the compactness of $\mathbb{Y}$, we get that the sequence $\{f_\sigma^{\epsilon,q}\}_{q\geq p}$ is uniformly bounded in $W^{1,p}((\mathbb{X},\mu),\mathbb{Y})$ as $\|f_\sigma^{\epsilon,q}\|_{W^{1,p}((\mathbb{X},\mu),\mathbb{Y})} \leq$

$M + \bar{\alpha}(\epsilon)$. Moreover, for $\dim(\mathbb{X}) < p \leq \infty$, we have from Morrey's inequality that:

$$|f_\sigma^{\epsilon,p}(x_1) - f_\sigma^{\epsilon,p}(x_2)| \leq \frac{2p \dim(\mathbb{X})}{p - \dim(\mathbb{X})} |x_1 - x_2|^{1 - \frac{\dim(\mathbb{X})}{p}} \, \||\nabla f_\sigma^{\epsilon,p}|\|_{L^p(\mathbb{X},\mu)}$$

$$\leq 2C \dim(\mathbb{X}) (1 + \dim(\mathbb{X})) |x_1 - x_2|^{\frac{1}{1+\dim(\mathbb{X})}} \bar{\alpha}(\epsilon),$$

where $C = \max\left\{1, \mathrm{diam}(\mathbb{X})^{\frac{\dim(\mathbb{X})}{1+\dim(\mathbb{X})}}\right\}$. It follows from the above that the sequence $\{f_\sigma^{\epsilon,p}\}_{p \in \mathbb{N}, p > \dim(\mathbb{X})}$ is also uniformly equicontinuous. Therefore, by the Arzelà-Ascoli Theorem [3], there exists a subsequence $\{f_\sigma^{\epsilon,p_j}\}_{j \in \mathbb{N}}$ that converges uniformly to a Lipschitz continuous $f_\sigma^{\epsilon,\infty}$. Moreover, from the monotonicity of minimum values, it follows that the Lipschitz constant $\mathrm{lip}(f_\sigma^{\epsilon,\infty}) = \||\nabla f_\sigma^{\epsilon,\infty}|\|_{L^\infty(\mathbb{X},\mu)} \leq \bar{\alpha}(\epsilon)$. We also have $\mathrm{lip}(f_\sigma^{\epsilon,\infty}) \geq \min_{\substack{f \in \mathrm{Lip}((\mathbb{X},\mu);\mathbb{Y}) \\ L_\sigma(f) \leq J_\sigma^*(\alpha)+\epsilon}} \mathrm{lip}(f) = \bar{\alpha}(\epsilon)$. Therefore, we have $\mathrm{lip}(f_\sigma^{\epsilon,\infty}) = \bar{\alpha}(\epsilon)$, and $\{f_\sigma^{\epsilon,p}\}_{p \in \mathbb{N}}$ converges uniformly (upto a subsequence) to a (global) minimizer $f^{\epsilon,\infty}$ of (10).

## G  Numerical analysis of classifier robustness

In this section, we provide numerical analysis to quantify a classifier's robustness against data perturbation for the classification problem discussed in Section 2 and Fig. 1 of the manuscript. Using the same setup explained in Section 2, we design our classifiers by constructing a graph $\mathcal{G} = (\mathcal{V}, \mathcal{E})$ with $n = 500$ randomly selected nodes by connecting each node to its 10 nearest neighbors. We compute the solution $\mathbf{v}^*$ to (9) for different values of the Lipschitz constant $\alpha \in (0, 100]$. We generate a nominal testing set of 1000 i.i.d. samples from $\sigma$, associate them with the closest node, and evaluate the nominal classification confidence of $\mathbf{v}^*$. Then, we perturb each testing data sample with $\delta \in \mathbb{R}^2$ with $\|\delta\|_2 = 0.05$ in the direction perpendicular to the closest edge, associate each perturbed data point with the closest node and evaluate the perturbed classification confidence. To measure the sensitivity of the designed classifier, we compute the norm of the difference between the nominal and the perturbed confidence, then appoint the sensitivity measure to the maximum value across all the testing data points. Fig. G.1(a) shows the plot of the sensitivity for each classifier designed using different Lipschitz bound $\alpha$, it can be seen that the sensitivity increases as we increase the Lipschitz bound up to $\alpha = 18$. Fig. G.1(b) shows the plot of the sensitivity for each classifier as a function of the classification confidence, we observe a tradeoff between classification performance and robustness to data perturbation seen by the monotonic increase of the sensitivity as a function of classification confidence, where improving classification performance comes at the expenses of robustness to data perturbation.

Figure G.1: For the classification problem discussed in Section 2 and Fig. 1 in the main manuscript, (a) shows the classifier's sensitivity to data perturbation as a function of the Lipschitz bound, the plot shows that sensitivity increases with the Lipschitz bound up to a certain value ($\alpha = 18$). (b) shows the tradeoff between performance and robustness, seen by the monotonic increase of the sensitivity as a function of classification confidence.

## Footnotes

[1]Given a measurable map $T : \mathbb{Z} \to \mathbb{Z}'$ and a probability measure $\sigma \in \mathcal{P}(\mathbb{Z})$, we let $T_\# \sigma$ denote the pushforward of $\sigma$ by the map $T$, where for any Borel measurable set $B \subset \mathbb{Z}'$ we have $T_\# \sigma(B) = \sigma(T^{-1}(B))$.

[2] For a countably additive measure $\nu$ that is absolutely continuous w.r.t. the Lebesgue measure, and any collection of nonempty sets $\{E_n\}_{N\in\mathbb{N}}$ with $E_{n+1} \subset E_n$ and $\lim_{n\to\infty} E_n = \emptyset$, we have $\lim_{n\to\infty} \nu(E_n) = 0$ [5].