[Reviews · NeurIPS 2020]

Review 1

Summary and Contributions: This paper studies the solution of loss minimization problem with Lipschitz constraints. The problem is mainly studied at a abstract level (assuming the map f is drawn from all Lipschitz maps rather than using a specific type of classifier). The main theorems in this paper give first order optimality conditions of this problem, in a similar manner as KKT conditions.

Strengths: The problem of Lipschitz constrained loss minimization is an important theoretical problem and related to many machine learning problems. The main theorems in this paper look technically sound and the connections to PDEs are interesting. Although the main theorems in this paper are abstract, the authors propose a discretization scheme that allows to apply the theorems to practical problems, but I am not sure how well it works for high dimensional problems as it requires N --> infinity. Figure 1 is very helpful for understanding the main idea of this paper.

Weaknesses: Lipschitz constant constrained models are usually not a good approach to obtain a robust model (see [1], the global Lipschitz constraint is usually too strong to be useful), and this is not the only approach for adversarially robust learning. A model can have large Lipschitz constant but can still be robust: for example, suppose we have a robust classifier f(x), we can add a small but high "frequency" sinusoidal signal to it: g(x)=f(x)+0.1sin(1000x). g(x) can still be robust if f(x) is relatively large so the sin() part can be ignored in zero-th order, but the Lipschitz constant of g(x) is large because the gradient of sin(1000x) is large. So the proposed theory has limitations that need to be discussed. Empirical evaluation of the proposed discretization method to solve real datasets are too weak. My understanding from the abstract is that the paper proposed a provably robust training scheme, yet it is not well demonstrated. Figure 2(c) 2(d) seems weird - even epsilon is very large, allowing a very large gap between the optimal loss, the Lipschitz constant is only reduced by 0.1 and still far from zero (unlike in the synthetic data in Figure 2(b). Also, Figure 2(e) and 2 (f) need to show a larger range of accuracy/confidence (for example, accuracy from 0.50 to 0.98 rather than 0.92 to 0.98).

Correctness: The method looks correct. For empirical study, the main text only has very limited information on the MNIST experiments. I also checked the supplementary and cannot find more details. So I do have concerns on reproducibility.

Clarity: The overall flow of this paper is clear.

Relation to Prior Work: I am not particular familiar with existing theory on solving Lipschitz constrained optimization problems so I am not sure if the discussions of previous works in this paper are sufficient.

Reproducibility: No

Additional Feedback: I am not particular familiar with existing theory on solving Lipschitz constrained optimization, but the main theory in this paper looks novel. The paper currently looks like unfinished and has some missing details, especially on how the proposed algorithm is applied to MNIST. Also it is worthwhile to study a few other datasets with different complexity, to show their "fundamental lower bound" (line 18). The provably robust training scheme needs to be demonstrated on a few small datasets. I believe this paper has the potential to become a very good paper, but currently it looks incomplete for this conference. [1] Huster, Todd, Cho-Yu Jason Chiang, and Ritu Chadha. "Limitations of the Lipschitz constant as a defense against adversarial examples." Joint European Conference on Machine Learning and Knowledge Discovery in Databases. Springer, Cham, 2018. -----------------------After author response--------------------- Thank you for providing the detailed response. Thanks for explaining about robustness and Lipschitz constant. However I am still not totally convinced. As mentioned in the response, Lipschitz constant is "targeting precisely the regions of *peak sensitivity* of the model". If the peak sensitivity appears in a region that is already always classified as the same label, a large Lipschitz constant does not change the prediction, so does not necessarily mean the model is not robust. It seems to me that it matters only near the decision boundaries, or maybe local Lipschitz constants make more sense. Since a small global Lipschitz constant is not a necessary condition for robustness, using Lipschitz regularization can be too strong and harmful, and that is one of my main concerns. Lipschitz constant has been discussed in a few early works on robustness such as [1][2], so this paper is definitive not the first paper to connect Lipschitz constant with robustness. I feel the graph-based algorithm is the most interesting part, but in the meanwhile I believe the mathematical formulation in this paper is unnecessarily complicated for many ML audience and the paper lacks a procedural algorithm description. It is still unclear to me exactly how the proposed algorithm is applied to MNIST (although the authors promised to add more details in the final version). If the paper claims "a provable training scheme" as a contribution, the authors should definitely improve their paper on that aspect. Overall, I think this paper is acceptable if the Lipschitz regularization optimization theory is novel enough, but honestly I think the proposed algorithm is not very clearly presented to me and the empirical evaluation needs to be enhanced. I have increased my score, however my confidence on this paper is relatively low, and I hope other reviewers or the AC can make a better judgement. [1] Moustapha Cisse et al. Parseval Networks: Improving Robustness to Adversarial Examples, ICML 2017. [2] Matthias Hein & Maksym Andriushchenko. Formal Guarantees on the Robustness of a Classifier against Adversarial Manipulation. NIPS 2017.


Review 2

Summary and Contributions: The authors consider two complementary approaches to provably robust learning. First they consider robust learning as minimization of a strictly convex loss function under a lipschitz constraint. They show the lagrangian of this minimization problem has a unique saddle point with the primal optimal solution adhering to the Lipschitz constraint, and leverage this insight to develop a provably lipschitz training scheme. Second, the authors consider the complementary problem where they seek to minimize the lipschitz constant with a loss-margin constraint. A similar approach is utilized to draw insights on the tradeoff between robustness and accuracy. ---------------------------- Updating after rebuttal: I'm not particularly satisfied by the author response, however I will keep my score as originally posted: I think this paper is above the acceptance threshold but marginally so. The presentation is slightly too formal for the standard ML/AI audience, and there are many details that I did not follow precisely, but I think there are enough worthwhile contributions here (namely the graph-based algorithm and a formal statement of the accuracy<->robustness tradeoff) for acceptance. I'm not confident enough in my score to be a strong advocate, but I am leaning towards accept.

Strengths: The authors provide a very formal theoretical treatment to a problem of much interest to the machine learning community. Applying classic theories of elliptic operators and PDE's is a novel perspective and can provide further theoretical evidence for the 'folklore' belief that accuracy is at odds with robustness. Section 2 provides insight into a novel model/training scheme that enforces a Lipschitz constraint a.e. (when optimized fully) and attains maximal accuracy in the limit. Section 3 provides the more interesting insight due to a very cute application of complementary slackness to demonstrate that there is indeed a trade-off between loss and Lipschitz constants. These theoretical claims are supported by an easy-to-understand example.

Weaknesses: My major complaints can be characterized into the following bullet points: - Robustness is argued only indirectly by way of Lipschitz constants. While the authors present a novel formulation (i.e., differing from the standard low-lipschitz=>robustness claims in the ML literature) of how controlling the lipschitz function of the classifier controls sensitivity to adversarial perturbations, this setting differs slightly from standard classification settings. In the standard classification setting, where labels are 1-hot vectors in R^dim(Y), a classifier typically returns as a label the argmax of the vector-valued function. Robustness then is usually considered as whether the argmax returns the right label, rather than a strictly-convex loss applied to the one-hot-label: this work incorporates 'confidence' into the robustness evaluation. - The applicability of the robust training scheme seems unlikely to scale to practical datasets, particularly those supported on high-dimensional domains. It seems like the accuracy would scale unfavorably unless the size of V scales exponentially with the dimension. - The example data distribution could be better chosen. As it stands, a classifier with (true) loss would require a Lipschitz constant that tends to infinity. A more standard/practical dataset would admit a perfect classifier that has a valid Lipschitz constant.

Correctness: The empirical methodology is correct. I am relatively unfamiliar with the mathematical preliminaries (elliptic operators/PDE's), so I am unable to thoroughly check the proofs of theoretical claims.

Clarity: The paper is clearly written and the remarks after theorem statements are very insightful and much appreciated. Mathematical preliminaries are provided for thinking about function spaces, but a pointer to or a brief discussion of the mathematical tools needed to derive the main theorems would be helpful.

Relation to Prior Work: Adequate discussion of the related work relevant work studying robustness and lipschitz regularization of ML models is provided.

Reproducibility: Yes

Additional Feedback: Acknowledging that this work is theoretical in nature, it is better for full reproducibility if code is provided along with the submission.


Review 3

Summary and Contributions: In this paper, the authors propose a graph-based learning framework, which could help train models with provable robustness. 1. Under the setting of Lipschitz constraint, the authors show that the saddle point of the Lagrangian is associated with a Poisson equation. Besides, the authors also introduce a graph-based discretization to solve the problem numerically. 2. Under the guaranteed bound on its loss, the authors show that the Lipschitz constant is tightly and inversely related to the loss. I think this paper has some contributions, can contains some novelty. But some flaws that cannot be ignored also appear, which will be listed in the Weaknesses Part. ========= After rebuttal After reading the response and the other reviews, I decide to keep my score unchanged.

Strengths: 1. The theorems in this paper are claimed clearly. At least, I can understand these theorems nearly without confusion. 2. I think this paper contains some novelty. For example, they connect provably robust learning with the classic theory of partial differential equations.

Weaknesses: 1. I think the conclusion "the Lipschitz constant is tightly and inversely related to the loss" is somehow trivial. Could the authors explain to us whether there are no theoretical explanations before, or you just use some novel methods to reach this conclusion? What are the differences between your conclusion and the others (e.g. different settings)? 2. Since two types of problems are considered here, could the authors further explain some more connections between these two problems except for their similar forms? 3. I think the problems this paper tries to solve are not motivated very well. Some more discussions could be added to better show the importance of this problem.

Correctness: Yes, I think the claims in this paper are correct.

Clarity: This paper should be polished carefully before publishment. It is a little bit hard to read due to some terms. For example, when claiming the novelty, the authors use "elliptic operators". But in the title and main text, they use "Laplace operators". It might be a little bit hard for beginners to understand these differences.

Relation to Prior Work: This paper shows a related work part. But I think some details should be added. For example, this paper says "This paper builds and extends upon these early studies.", but with no further discussion on HOW.

Reproducibility: Yes

Additional Feedback: Overall, I think this paper tells a good story, and the merits overweight the flaws. And I think I would give a higher score with better writing.


Review 4

Summary and Contributions: This paper studied the empirical risk minimization of Lipschitz continuous functions, where two formulations are proposed by either bounding the Lipschitz constant from above as a constraint or searching for the function with minimum Lipschitz constant such that the empirical risk does not violate the optimal risk by a margin.

Strengths: 1. The Lipschitz-constrained loss minimization is formulated with rigorous math notations. The KKT conditions (Thm 2.1& 3.1) of two formulations are well analyzed. 2. The discretized versions (eq. 7 & eq. 14) are interesting and allow us to understand the minimization problems from a pure optimization perspective. 3. The second formulation (eq. 14) offers an insight on the tradeoff between performance and robustness.

Weaknesses: 1. The mathematical rigorousness renders the reading much more difficult which I don't think it's necessary for a ML paper. 2. The connection between the discretized versions and Laplace regularization for semi-supervised learning should be made more clear. It was not clear for me why it is relevant to consider the discretized versions. 3. Since the loss minimization problems are wrt functions, the problems are convex, but in practice, we don't optimize high-dimensional functions directly; instead, a parameterization of the function is introduced and the loss minimization wrt the parameters is not necessarily convex. Can the results of this paper be extended to those cases? Are these results applicable for neural networks (since the main applications of Lip constraints are there)?

Correctness: The formulation and theorems look correct to me, but I didn't check the proofs.

Clarity: L194: "sensitivity to adversarial perturbations" -- How can I see that? It seems the top right is the best where n and alpha are both high. L255: Could you extend the key insight from Thm 3.1? Not clear why perf and robust cannot be achieved at the same time.

Relation to Prior Work: The related work section mentioned several prior works on Lip constraints for neural networks, but a more extensive review could be given, e.g. to add a review on the relationship between Lip constraint and semi-supervised learning, and Lip constraint to GAN.

Reproducibility: No

Additional Feedback: # After rebuttal Due to the space limit, the authors didn't completely address my concerns. The analysis in the paper is interesting in the sense of adversarial robustness while I don't find this paper particularly impactful as the results may not be able to extend to non-convex cases.

[Author Response · NeurIPS 2020]

We thank all the reviewers for their valuable comments. We would be happy to revise our paper using their suggestions.

**Reviewer 1: Lipschitz constant and adversarial robustness.** While we agree that different approaches can be used
to improve adversarial robustness, optimizing the Lipschitz constant of a model is shown to tightly bound the *worst-case*
error due to bounded input perturbation (See Fig 1(e) below from our experiments; our SI and *Fazlyab et al., NeurIPS*
*2019*). The Lipschitz constraint is particularly effective at targeting precisely the regions of peak sensitivity of the
model, and our approach thereby provides provable guarantees for the trained model against adversarial perturbations.

We respectfully disagree with the statement that a "model can have large Lipschitz constant but can still be robust". In
the provided example, if $g$ achieves a higher performance than $f$, then the model $g$ lacks robustness. In fact, a very
small perturbation (of the order of $1/1000$) can degrade the performance of $g$ at all testing points close to the decision
boundaries (which are, in the absence of perturbation, correctly classified due to the high-frequency component). If $g$
achieves the same performance as $f$, then $g$ contains an unnecessary high-frequency term. In fact, because our loss
function is strictly convex, it admits a unique minimizer, and high-frequency components are present in our models only
if they improve the overall performance (our Laplacian approach favors smooth solutions). This analysis is compatible
with the result that the Lipschitz constant provides a worst-case bound on the robustness of a model to perturbations.

Finally, despite its title, the reference cited by Reviewer 1 does not claim that bounding the Lipschitz constant is not a
good approach to obtain a robust model. Rather, it argues in favor of our work: it claims that the existing methods to
compute the Lipschitz constant had limitations, and that overcoming them would lead to adversarially robust models.

**Reviewers 1 and 4: Empirical validations, figures, and reproducibility.** We thank the reviewers for pointing out
these weaknesses. We recreated our figures (below; which now align with Reviewer 1's intuition). We can now provide
more MNIST implementation details, upload our code to GitHub, and further demonstrate our robust training scheme.

**Reviewer 2: Confidence vs accuracy.** Confidence and accuracy are directly related (Figure 1(d)). Confidence is a
smoother metric for optimization and encoding the Lipschitz constraint, and is readily mapped onto one-hot vectors.

**Reviewer 2: Scalability.** We are able to successfully demonstrate our approach using the MNIST dataset. While
additional studies are needed to investigate scalability, our numerical studies show that our performance increases
rapidly with the number of vertices (model complexity), with an accuracy of $96\%$ with only 10000 vertices (Figure 1(c)).
For comparison, our study shows that neural networks with the same number of parameters achieve only $68\%$ accuracy.

**Reviewer 2: Dataset.** We choose the checkerboard dataset precisely because it requires a Lipschitz constant that tends
to infinity, so as to compare the performance over a broad range. We use the MNIST dataset for a more realistic study.

**Reviewer 3: Conclusion and novelty.** To the best of our knowledge, our results are the first to prove that a fundamental
tradeoff exists between Lipschitz constant and accuracy. Prior works (cited) provide empirical or less general results.
Our paper contributes both new results to the literature to address the immediate questions on tradeoffs, and a novel way
of looking at the learning problem (connections to PDE) that has broader scope and will pave the way for future works.

**Reviewer 3: Connections between problems and motivation.** The first problem generates models with minimal loss
and desired robustness. The second problem fixes the performance (loss) and improves the robustness of a model. While
the first problem yields a training scheme, the second is used to prove a fundamental tradeoff between performance and
robustness. We have drawn motivation for our work from prior works on adversarial attacks that have demonstrated a
glaring problem of robustness of machine learning models and a need for adversarially robust training.

**Reviewer 4: Discretized models and neural networks.** The discretized versions of the problems offer a way to
construct an implementable training scheme with robustness guarantees. Implementable models are obtained by various
discretizations of the original infinite-dimensional loss minimization problem. We have considered a graph discretization
that directly inherits the convexity of the infinite-dimensional problem. As the reviewer points out, other function
space parametrizations, such as neural networks, result in non-convex finite-dimensional optimization problems. Our
infinite-dimensional analysis holds regardless of parametrization and characterizes the solution that particular models
approximate, while our graph-based design offers a novel and alternative scheme to construct provably-robust models.

*Figure 1: Figure (a) updates Fig. 2c in the paper. Figures (b)-(e) complement our study of the MNIST dataset by looking at the relations between accuracy and Lipschitz, accuracy and complexity, accuracy and confidence, adversarial robustness and Lipschitz.*

[Meta-Review · NeurIPS 2020]

Reviewers found the idea of the paper interesting but there were some concerns about the practicality of the approach. Authors should carefully address the comments raised by the reviewers.